# A View From Somewhere:
# Human-Centric Face Representations

**Jerone T. A. Andrews**[*]
Sony AI, Tokyo

**Przemysław Joniak**[†]
University of Tokyo, Tokyo

**Alice Xiang**
Sony AI, New York

## Abstract

Few datasets contain self-identified sensitive attributes, inferring attributes risks introducing additional biases, and collecting attributes can carry legal risks. Besides, categorical labels can fail to reflect the continuous nature of human phenotypic diversity, making it difficult to compare the similarity between same-labeled faces. To address these issues, we present A View From Somewhere (AVFS)—a dataset of 638,180 human judgments of face similarity.[1] We demonstrate the utility of AVFS for learning a continuous, low-dimensional embedding space aligned with human perception. Our embedding space, induced under a novel conditional framework, not only enables the accurate prediction of face similarity, but also provides a human-interpretable decomposition of the dimensions used in the human-decision making process, and the importance distinct annotators place on each dimension. We additionally show the practicality of the dimensions for collecting continuous attributes, performing classification, and comparing dataset attribute disparities.

## 1 Introduction

The canonical approach to evaluating human-centric image dataset diversity is based on demographic attribute labels. Many equate diversity with parity across the subgroup distributions (Kay et al., 2015; Schwemmer et al., 2020), presupposing access to demographically labeled samples. However, most datasets are web scraped, lacking ground-truth information about image subjects (Andrews et al., 2023). Moreover, data protection legislation considers demographic attributes to be personal information and limits their collection and use (Andrus et al., 2021; 2020).

Even when demographic labels are known, evaluating diversity based on subgroup counts fails to reflect the continuous nature of human phenotypic diversity (e.g., skin tone is often reduced to "light" vs. "dark"). Further, even within the same subpopulation, image subjects exhibit certain traits to a greater or lesser extent than others (Becerra-Riera et al., 2019; Carcagnì et al., 2015; Feliciano, 2016).

When labels are unknown, researchers typically choose certain attributes they consider to be relevant for human diversity and use human annotators to infer them (Karkkainen & Joo, 2021; Wang et al., 2019). Inferring labels, however, is difficult, especially for nebulous social constructs, e.g., race and gender (Hanna et al., 2020; Keyes, 2018), and can introduce additional biases (Freeman et al., 2011). Beyond the inclusion of derogatory categories (Koch et al., 2021; Birhane & Prabhu, 2021; Crawford & Paglen, 2019), label taxonomies often do not permit multi-group membership, resulting in the erasure of, e.g., multi-ethnic individuals (Robinson et al., 2020; Karkkainen & Joo, 2021). Significantly, discrepancies between inferred and self-identified attributes can induce psychological distress by invalidating an individual's self-image (Campbell & Troyer, 2007; Roth, 2016).

In this work, we avoid problematic semantic labels altogether and propose to learn a face embedding space aligned with human perception. To do so, we introduce A View From Somewhere (AVFS)—a dataset of 638,180 face similarity judgments over 4,921 faces. Each judgment corresponds to the odd-one-out (i.e., least similar) face in a triplet of faces and is accompanied by both the identifier and demographic attributes of the annotator who made the judgment. Our embedding space, induced under a novel conditional framework, not only enables the accurate prediction of face similarity, but also

---

[*]Correspondence to `jerone.andrews@sony.com`.
[†]Work done while the author was an intern at Sony AI, Tokyo.
[1]Code and data may be found at `https://github.com/SonyAI/a_view_from_somewhere`.

provides a human-interpretable decomposition of the dimensions used in the human decision-making process, as well as the importance distinct annotators place on each dimension. We demonstrate that the individual embedding dimensions (1) are related to concepts of gender, ethnicity, age, as well as face and hair morphology; and (2) can be used to collect continuous attributes, perform classification, and compare dataset attribute disparities. We further show that annotators are influenced by their sociocultural backgrounds, underscoring the need for diverse annotator groups to mitigate bias.

## 2 RELATED WORK

**Similarity.** The human mind is conjectured to have, "a considerable investment in similarity" (Medin et al., 1993). When two objects are compared they mutually constrain the set of features that are activated in the human mind (Markman, 1996)—i.e., features are dynamically discovered and aligned based on what is being compared. Alignment is contended to be central to similarity comparisons. Shanon (1988) goes as far as arguing that similarity is not a function of features, but that the features themselves are a function of the similarity comparison.

**Contextual similarity.** Human perception of similarity can vary with respect to context (Roth & Shoben, 1983). Context makes salient context-related properties and the extent to which objects being compared share these properties (Medin et al., 1993; Markman & Gentner, 2005; Goodman, 1972). For example, Barsalou (1987) found that "snakes" and "raccoons" were judged to be more similar when placed in the context of pets than when no context was provided. The odd-one-out similarity task (Zheng et al., 2019) used in this work also provides context. By varying the context (i.e., the third object) in which two objects are experienced, it is possible to uncover different features that contribute to their pairwise similarity. This is important as there are an uncountable number of ways in which two objects may be similar (Love & Roads, 2021).

**Psychological embeddings.** Multidimensional scaling (MDS) is often used to learn psychological embeddings from human similarity judgments (Zheng et al., 2019; Roads & Love, 2021; Dima et al., 2022; Josephs et al., 2021). As MDS approaches cannot embed images outside of the training set, researchers have used pretrained models as feature extractors (Sanders & Nosofsky, 2020; Peterson et al., 2018; Attarian et al., 2020), which can introduce unwanted implicit biases (Krishnakumar et al., 2021; Steed & Caliskan, 2021). Moreover, previous approaches ignore inter- and intra-annotator variability. By contrast, our conditional model is trained end-to-end and can embed any arbitrary face from the perspective of a specific annotator.

**Face datasets.** Most face datasets are semantically labeled images, created for the purposes of identity and attribute recognition (Karkkainen & Joo, 2021; Liu et al., 2015; Huang et al., 2008; Cao et al., 2018). An implicit assumption is that semantic similarity is equivalent to visual similarity (Deselaers & Ferrari, 2011). However, many semantic categories are functional (Rosch, 1975; Rothbart & Taylor, 1992), i.e., unconstrained by visual features such as shape, color, and material. Moreover, semantic labels often only indicate the presence or absence of an attribute, as opposed to its magnitude, making it difficult to compare the similarity between same-labeled samples. While face similarity datasets exist, the judgments narrowly pertain to identity (McCauley et al., 2021; Sadovnik et al., 2018; Somai & Hancock, 2021) and expression similarity (Vemulapalli & Agarwala, 2019).

**Annotator positionality.** Semantic categorization by annotators not only depends on the image subject, but also on extrinsic contextual cues (Freeman et al., 2011) and the annotators' sociocultural backgrounds (Segall et al., 1966). Despite this, annotator positionality is rarely discussed in computer vision (Chen & Joo, 2021; Zhao et al., 2021; Denton et al., 2021); "only five publications [from 113] provided any [annotator] demographic information" (Scheuerman et al., 2021). To our knowledge, AVFS represents the first human-centric vision dataset, where each annotation is associated with the annotator who created it and their demographics, permitting the study of annotator bias.

## 3 A VIEW FROM SOMEWHERE DATASET

To learn a face embedding space aligned with human perception, we collect AVFS—a dataset of odd-one-out similarity judgments collected from humans. An odd-one-out judgment corresponds to the least similar face in a triplet of faces, representing a three-alternative forced choice (3AFC) task. AVFS dataset documentation can be found in Appendix C.

**Face image stimulus set.** 4,921 near-frontal faces with limited eye occlusions and an apparent age $> 19$ years old were sampled from the CC-BY licensed FFHQ (Karras et al., 2019) dataset. The subset was obtained by (1) splitting FFHQ into 56 partitions based on inferred intersectional group labels; and (2) randomly sampling from each partition with equal probability. Ethnicity was estimated using a FairFace model (Karkkainen & Joo, 2021); and binary gender expression and age group labels were obtained from FFHQ-Aging crowdsourced human annotations (Or-El et al., 2020).

**3AFC similarity judgments.** AVFS contains 638,180 quality-controlled triplets over 4,921 faces, representing 0.003% of all possible triplets. To focus judgments on intrinsic face-varying features, for each presented triplet, annotators were instructed to choose the person that looks least similar to the two other people, while ignoring differences in pose, expression, lighting, accessories, background, and objects. Each AVFS triplet is labeled with a judgment, as well as the identifier of the annotator who made the judgment and the annotator's self-reported age, nationality, ancestry, and gender identity. As in previous non-facial odd-one-out datasets (Josephs et al., 2021; Hebart et al., 2022; Dima et al., 2022), there is a single judgment per triplet. Quality was controlled by excluding judgments from annotators who provided overly fast, deterministic, or incomplete responses. In total, 1,645 annotators contributed to AVFS via Amazon Mechanical Turk (AMT) and provided consent to use their study data. Compensation was 15 USD per hour.

**3AFC task rationale.** Let $\mathbf{x} \in \mathcal{X}$ and $\mathrm{sim} : (\mathbf{x}_i, \mathbf{x}_j) \to \mathrm{sim}(i, j) \in \mathbb{R}$ denote a face image and a similarity function, respectively. Our motivation for collecting AVFS is fourfold. Most significantly, the odd-one-out task does not require an annotator to explicitly categorize people. Second, for a triplet $(\mathbf{x}_i, \mathbf{x}_j, \mathbf{x}_k)$, repeatedly varying $\mathbf{x}_k$ permits the identification of the relevant dimensions that contribute to $\mathrm{sim}(i, j)$. That is, w.l.o.g., $\mathbf{x}_k$ provides context for which $\mathrm{sim}(i, j)$ is determined, making the task easier than explicit pairwise similarity tasks (i.e., "Is $\mathbf{x}_i$ similar to $\mathbf{x}_j$?"). This is because it is not always apparent to an annotator which dimensions are relevant when determining $\mathrm{sim}(i, j)$, especially when $\mathbf{x}_i$ and $\mathbf{x}_j$ are perceptually different. Third, there is no need to prespecify attribute lists hypothesized as relevant for comparison (e.g., "Is $\mathbf{x}_i$ older than $\mathbf{x}_j$?"). The odd-one-task implicitly encodes salient attributes that are used to determine similarity. Finally, compared to 2AFC judgments for triplets composed of an anchor (i.e., reference point), $\mathbf{x}_a$, a positive, $\mathbf{x}_p$, and a negative, $\mathbf{x}_n$, 3AFC judgments naturally provide more information. 3AFC triplets require an annotator to determine $\{\mathrm{sim}(i, j), \mathrm{sim}(i, k), \mathrm{sim}(j, k)\}$, whereas 2AFC triplets with anchors only necessitate the evaluation of $\{\mathrm{sim}(a, p), \mathrm{sim}(a, n)\}$.

## 4 MODEL OF CONDITIONAL DECISION-MAKING

Zheng et al. (2019) developed an MDS approach for learning continuous, non-negative, sparse embeddings from odd-one-out judgments. The approach has been shown to offer an intepretable window into the dimensions in the human mind of object categories (Hebart et al., 2020), human actions (Dima et al., 2022), and reachspace environments (Josephs et al., 2021) such that dimensional values indicate feature magnitude. The method is based on three assumptions: (1) embeddings can be learned solely from odd-one-out judgments; (2) judgments are a function of $\{\mathrm{sim}(i, j), \mathrm{sim}(i, k), \mathrm{sim}(j, k)\}$; and (3) judgments are stochastic, where the probability of selecting $x_k$ is

$$p(k) \propto \exp(\mathrm{sim}(i, j)). \tag{1}$$

**Conditional similarity.** At a high-level, we want to apply Zheng et al. (2019)'s MDS approach to learn face embeddings. However, the method (1) cannot embed data outside of the training set, limiting its utility; and (2) pools all judgments, disregarding intra- and inter-annotator stochasticity. Therefore, we instead propose to learn a conditional convolutional neural net (convnet).

Let $\{(\{\mathbf{x}_{i\ell}, \mathbf{x}_{j\ell}, \mathbf{x}_{k\ell}\}, k\ell, a)\}_{\ell=1}^{n}$ denote a training set of $n$ (triplet, judgment, annotator) tuples, where $a \in \mathcal{A}$. To simplify notation, we assume that judgments always correspond to index $k\ell$. Suppose $f : \mathbf{x} \mapsto f(\mathbf{x}) = \mathbf{w} \in \mathbb{R}^d$ is a convnet, parametrized by $\mathbf{\Theta}$, where $\mathbf{w} \in \mathbb{R}^d$ is an embedding of $\mathbf{x}$. In contrast to Zheng et al. (2019), we model the probability of annotator $a$ selecting $k\ell$ as

$$p(k\ell \mid a) \propto \exp(\mathrm{sim}_a(i\ell, j\ell)), \tag{2}$$

where $\mathrm{sim}_a(\cdot, \cdot)$ denotes the internal similarity function of $a$. Given two images $\mathbf{x}_i$ and $\mathbf{x}_j$, we define their similarity according to $a$ as:

$$\mathrm{sim}_a(i, j) = (\sigma(\boldsymbol{\phi}_a) \odot \mathrm{ReLU}(\mathbf{w}_i))^{\top} \cdot (\sigma(\boldsymbol{\phi}_a) \odot \mathrm{ReLU}(\mathbf{w}_j)), \tag{3}$$

where $\sigma(\cdot)$ is the sigmoid function and $\sigma(\phi_a) \in [0,1]^d$ is a mask associated with $a$. Each mask plays the role of an element-wise gating function, encoding the importance $a$ places on each of the $d$ embedding dimensions when determining similarity. Conditioning prediction on $a$ induces subspaces that encode each annotator's notion of similarity, permitting us to study whether per-dimensional importance weights differ between $a \in \mathcal{A}$. Veit et al. (2017) employed a similar procedure to learn subspaces which encode different notions of similarity (e.g., font style, character type).

**Conditional loss.** We denote by $\Phi = [\phi_1^\top, \ldots, \phi_{|\mathcal{A}|}^\top] \in \mathbb{R}^{d \times |\mathcal{A}|}$ a trainable weight matrix, where each column vector $\phi_a^\top$ corresponds to annotator $a$'s mask prior to applying $\sigma(\cdot)$. In our conditional framework, we jointly optimize $\Theta$ and $\Phi$ by minimizing:

$$-\sum_\ell \log\left[\hat{p}(k\ell \mid a)\right] + \alpha_1 \sum_i \|\text{ReLU}(\mathbf{w}_i)\|_1 + \alpha_2 \sum_{ij} \text{ReLU}(-\mathbf{w}_i)_j + \alpha_3 \sum_a \|\phi_a\|_2^2, \quad (4)$$

where

$$\hat{p}(k\ell \mid a) = \frac{\exp(\text{sim}_a(i\ell, j\ell))}{\exp(\text{sim}_a(i\ell, j\ell)) + \exp(\text{sim}_a(i\ell, k\ell)) + \exp(\text{sim}_a(j\ell, k\ell))} \quad (5)$$

is the predicted probability that $k\ell$ is the odd-one-out conditioned on $a$. The first term in Equation (4) encourages similar face pairs to result in large dot products. The second term (modulated by $\alpha_1 \in \mathbb{R}$) promotes sparsity. The third term (modulated by $\alpha_2 \in \mathbb{R}$) supports interpretability by penalizing negative values. The last term (modulated by $\alpha_3 \in \mathbb{R}$) penalizes large weights.

Our conditional decision-making framework is applicable to any task that involves mapping from inputs to decisions made by humans; it only requires record-keeping during data collection such that each annotation is associated with the annotator who generated it.

**Implementation.** When $\alpha_3 = 0$ and $\sigma(\phi_a) = [1, \ldots, 1]$, $\forall a$, Equation (4) corresponds to the unconditional MDS objective proposed by Zheng et al. (2019). We refer to unconditional and conditional models trained on AVFS as AVFS-U and AVFS-C, respectively, and conditional models whose masks are learned post hoc as AVFS-CPH. An AVFS-CPH model uses a fixed unconditional model (i.e., AVFS-U) as a feature extractor to obtain face embeddings such that only $\Phi$ is trainable.

AVFS models have ResNet18 (He et al., 2016) architectures and output 128-dimensional embeddings. We use the Adam (Kingma & Ba, 2014) optimizer with default parameters, reserving 10% of AVFS for validation. Based on grid search, we empirically set $\alpha_1 = 0.00005$ and $\alpha_2 = 0.01$. For AVFS-C and AVFS-CPH, we additionally set $\alpha_3 = 0.00001$. Across independent runs, we find that only a fraction of the 128 dimensions are needed and individual dimensions are reproducible. Post-optimization, we remove dimensions with maximal values (over the stimulus set) close to zero. For AVFS-U, this results in 22 dimensions (61.9% validation accuracy). Note that we observe that 8/22 dimensions have a Pearson's $r$ correlation $> 0.9$ with another dimension. Refer to Appendix A for further implementation details.

## 5 EXPERIMENTS

The aim of this work is to learn a face embedding space aligned with human perception. To highlight the value of AVFS, we evaluate the performance of AVFS embeddings and conditional masks for predicting face similarity judgments, as well as whether they reveal (1) a human-interpretable decomposition of the dimensions used in the human decision-making process; and (2) the importance distinct annotators place on each dimension. We further assess the practicality of the dimensions for collecting continuous attributes, classification, and comparing dataset attribute disparities.

**Embedding methods.** Where appropriate, for comparison, we consider embeddings extracted from face identity verification and face attribute recognition models: CASIA-WebFace (Yi et al., 2014) is a face verification model trained with an ArcFace (Deng et al., 2019) loss on images from 11K identities; CelebA (Liu et al., 2015) is a face attribute model trained to predict 40 binary attribute labels (e.g., "pale skin", "young", "male"); and FairFace (Karkkainen & Joo, 2021) and FFHQ are face attribute models trained to predict gender, age, and ethnicity. All models have ResNet18 architectures and output 128-dimensional embeddings (prior to the classification layer). All baseline embeddings are normalized to unit-length, where the dot product of two embeddings determines their similarity. Additional details on the datasets and baselines are presented in Appendix B.

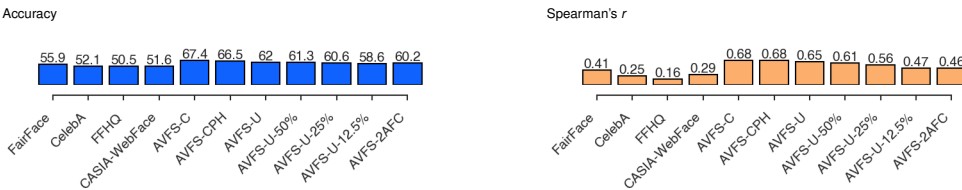

Figure 1: **Predicting similarity (same stimuli).** Accuracy over 24,060 judgments and Spearman's $r$ between the entropy in human- and model-generated triplet odd-one-out probabilities. Even when trained on a fraction of AVFS judgments (i.e., 12.5%), the learned AVFS-U-12.5% model still outperforms the baselines. Additional results are presented in Appendix A.

## 5.1 SAME STIMULI

**Predicting similarity.** We analyze whether our model results in stimulus set embeddings such that we are able to predict human 3AFC judgments not observed during learning. Using images from the stimulus set of 4,921 faces, we generate 1,000 novel triplets and collect 22–25 unique judgments per triplet (24,060 judgments) on AMT. As we have 22–25 judgments per triplet, we can reliably estimate odd-one-out probabilities. The Bayes optimal classifier accuracy corresponds to the best possible accuracy any model could achieve given the stochasticity in the judgments. The classifier makes the most probable prediction, i.e., the majority judgment. Thus, its accuracy is equal to the mean majority judgment probability over the 1,000 triplets, corresponding to 65.5±1% (95% CI).

In addition to accuracy, we report Spearman's $r$ correlation between the entropy in human- and model-generated triplet odd-one-out probabilities. Human-generated odd-one-out probabilities are of the form: $(n_i, n_j, n_k)/n$, where, without loss of generality, $n_k/n$ corresponds to the fraction of $n$ odd-one-out votes for $k$. Results are shown in Figure 1. We observe that AVFS models outperform the baselines with respect to accuracy, while better reflecting human perceptual judgment uncertainty (i.e., entropy correlation). This holds as we vary the number of AVFS training triplets, for example, to as few as 12.5% (72K) as evidenced by the AVFS-U-12.5% model. Further, our learned annotator-specific dimensional importance weights generalize, resulting in a predictive accuracy at or above the Bayes classifier's. As hypothesized (Section 3), transforming AVFS into 2AFC triplets (i.e., AVFS-2AFC in Figure 1) decreases performance due to a reduction in information.[2]

**Dimensional interpretability.** Following Hebart et al. (2020), we qualitatively assess whether learning on AVFS results in a human-interpretable decomposition of the dimensions used in the human decision-making process. Let $\mathbf{x}$ and $\text{ReLU}(\mathbf{w}) \in \mathbb{R}^{22}$ denote a stimulus set image and its AVFS-U model embedding, respectively. We denote by $\dim_j$ and $\%\text{tile}_j^q$ the $j^{\text{th}}$ dimension and percentile of $\dim_j$ stimulus set embedding values, respectively. Suppose $\text{grid}_j$ is a $5 \times 100$ face grid. Column $q \in [100, \ldots, 1]$ of $\text{grid}_j$ contains 5 stimulus set faces with the highest $\dim_j$ values, satisfying $\%\text{tile}_j^{q-1} \leq \text{ReLU}(\mathbf{w})_j < \%\text{tile}_j^q$. Thus, from left to right, $\text{grid}_j$ displays stimulus set faces sorted in descending order based on their $\dim_j$ value. We task annotators with writing 1–3 $\text{grid}_j$ descriptions. After controlling for quality, we obtain 23–58 labels (708 labels) for each $j$ and categorize the label responses into 35 topics.

Figure 2 evidences a clear relationship between the top 3 $\dim_j$ topics and $\dim_j$ labels generated using CelebA and FairFace models. Across the 22 dimensions, we note the materialization of individual dimensions coinciding with commonly collected demographic group labels, i.e., "male", "female", "black", "white", "east asian", "south asian", "elderly", and "young". In addition, separate dimensions surface for face and hair morphology, i.e., "wide face", "long face", "smiling expression", "neutral expression", "balding", and "facial hair". Significantly, these dimensions, which largely explain the variation in human judgment behavior, are learned without training on semantic labels.

## 5.2 NOVEL STIMULI

**Predicting similarity.** We evaluate whether AVFS models transfer to novel stimuli by sampling 56 novel face images from FFHQ (not contained in the stimulus set). We then generate all $\binom{56}{3} = 27,720$

---

[2]AVFS-2AFC was trained using a triplet margin with a distance swap loss (Balntas et al., 2016).

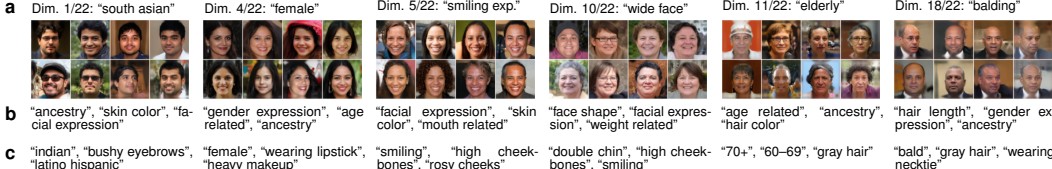

Figure 2: **Dimensional interpretability.** (a) Our interpretation of an AVFS-U dimension with the highest scoring stimulus set images. (b) Highest frequency human-generated topic labels. (c) CelebA and FairFace model-generated labels, exhibiting a clear correspondence with human-generated topics. Face images in this figure have been replaced by synthetic StyleGAN3 (Karras et al., 2021) images for privacy reasons. Additional dimensions are presented in the dataset documentation (Appendix C).

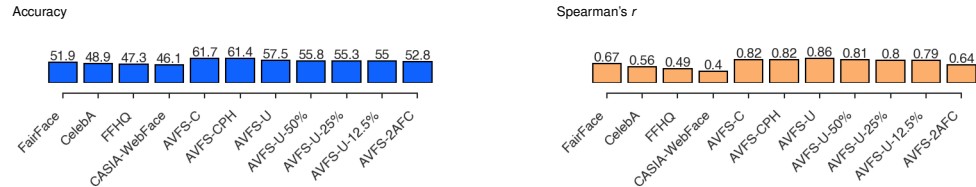

Figure 3: **Predicting similarity (novel stimuli).** Accuracy over 80,300 judgments and Spearman's $r$ between the entropy in human- and model-generated similarity matrices. Our annotator-specific masks (i.e., AVFS-C and AVFS-CPH models) generalize to triplets composed entirely of novel faces, resulting in increased predictive accuracy. Additional results are presented in Appendix A.

possible triplets, collecting 2–3 unique judgments per triplet (80,300 judgments) on AMT. We report accuracy, as well as Spearman's $r$ between the strictly upper triangular model- and human-generated similarity matrices. Entry $(i, j)$ in the human-generated similarity matrix corresponds to the fraction of triplets containing $(i, j)$, where neither was judged as the odd-one-out; and $(i, j)$ in a model-generated similarity matrix corresponds to the mean $\hat{p}(i, j)$ over all triplets containing $(i, j)$.

Results are shown in Figure 3. We see that AVFS embedding spaces and the learned annotator-specific masks generalize to triplets composed entirely of novel faces. This non-trivially demonstrates that AVFS embedding spaces are more closely aligned with the human mental representational space of faces, than embedding spaces induced by training on semantic labels.

To quantify the number of AVFS-U dimensions required to represent a face while preserving performance, we follow Hebart et al. (2020)'s dimension-elimination approach. We iteratively zero out the lowest value per face embedding until a single nonzero dimension remains. This is done per embedding, thus the same dimension is not necessarily zeroed out from all embeddings during an iteration. To obtain 95–99% of the predictive accuracy we require 6–13 dimensions, whereas to explain 95–99% of the variance in the similarity matrix we require 15–22 dimensions. This shows that (1) humans utilize a larger number of dimensions to represent the global similarity structure of faces than for determining individual 3AFC judgments; and (2) similarity is context-dependent (i.e., dynamic), which is not accounted for when representing faces using categorical feature vectors.

**Annotator positionality.** As shown via our conditional framework, knowledge of the annotator determining similarity assists in informing the outcome. However, annotators are often framed as interchangeable (Malevé, 2020; Chancellor et al., 2019). To assess the validity of this assumption, we randomly swap the annotator associated with each of the collected 80,300 judgments and recompute the AVFS-CPH model accuracy. Repeating this process 100 times results in a performance drop from 61.7% to 52.8% ± 0.02% (95% CI), evidencing that annotator masks, and hence annotators, are not arbitrarily interchangeable.

Another interesting question relates to whether an annotator's sociocultural background influences their decision making. To evaluate this, we create datasets $\{(\sigma(\phi_a^\top), y)\}$, where $\sigma(\phi_a^\top)$ and $y \in \mathcal{Y}$ are annotator $a$'s learned mask and self-identified demographic attribute, respectively. For a particular annotator attribute (e.g., nationality), we limit the dataset to annotators who contributed $\geq 200$ judgments and belong to one of the two largest groups with respect to the attribute (e.g.,

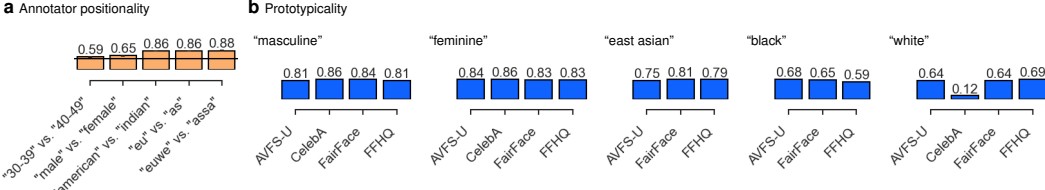

Figure 4: (a) **Annotator positionality.** Average linear SVM AUC when trained to discriminate between AVFS-CPH annotator-specific masks from different demographic groups. None of the confidence intervals overlap with chance performance (horizontal black line). Note the abbreviations: "europe" ("eu"), "asia" ("as"), "western europe" ("euwe"), and "southern asia" ("assa"). (b) **Prototypicality.** Spearman's $r$ between human ratings of face typicality and model-generated ratings, evidencing that social category typicality manifests in the AVFS-U model's embedding dimensions.

$\mathcal{Y} = \{$"american", "indian"$\}$). Using 10-fold cross-validation, we train linear SVMs (Hearst et al., 1998) with balanced class weights to predict $y$ from $\sigma(\phi_a^\top)$. Figure 4a shows that for each attribute none of the AUC confidence intervals include chance performance. Significantly, we are able to discriminate between nationality-, regional ancestry-, and subregional ancestry-based annotator group masks with high probability (86–88%). Thus, to mitigate bias, diverse annotator groups are required.

**Continuous attribute collection.** Following Hebart et al. (2020)'s evaluation protocol, we quantitatively validate whether $\text{grid}_j$ ($\forall j$)—defined in Section 5.1—can be used to collect continuous attribute values for novel faces from human annotators. We task annotators with placing a novel face $\mathbf{x}$ above a single column $q$ of $\text{grid}_j$. Placement is based on the similarity between $\mathbf{x}$ and the faces contained in each column $q$ of $\text{grid}_j$. As annotators are not primed with the meaning of $\text{grid}_j$, accurate placement indicates that the ordering of $\dim_j$ is visually meaningful.

Let $\mu_j^q$ denote the mean $\dim_j$ embedding value over the stimulus set whose $\dim_j$ value satisfies $\%\text{tile}_j^{q-1} \leq \text{ReLU}(\mathbf{w})_j < \%\text{tile}_j^q$. We sample 20 novel faces from FFHQ (not contained in the stimulus set). $\forall (\mathbf{x}, \text{grid}_j)$ and collect 20 unique judgments per face (8,800 judgments). $\forall \mathbf{x}$, we create a human-generated embedding of the form: $\left[\frac{1}{20}\sum_{n=1}^{20}\mu_1^{q_n}, \ldots, \frac{1}{20}\sum_{n=1}^{20}\mu_{22}^{q_n}\right] \in \mathbb{R}^{22}$, where $q_n \in [100, \ldots, 1]$ denotes the $n^{\text{th}}$ annotator's column choice $q$. We then generate all $\binom{20}{3}$ possible triplets to create a human-generated similarity matrix $\mathbf{K} \in \mathbb{R}^{20 \times 20}$. Entry $(i, j)$ corresponds to the mean $\hat{p}(i, j)$ over all triplets containing $(i, j)$ with the human-generated embeddings. Using model-generated embeddings, we create a model-generated matrix, $\hat{\mathbf{K}} \in \mathbb{R}^{20 \times 20}$, in the same way.

Spearman's $r$ correlation between the strictly upper triangular elements of $\mathbf{K}$ and $\hat{\mathbf{K}}$ is 0.83 and 0.86 for AVFS-U and AVFS-CPH embeddings, respectively. This shows that (1) dimensional values correspond to feature magnitude; and (2) image grids can be used to directly collect continuous attribute values, sidestepping the limits of semantic category definitions (Keyes, 2018; Benthall & Haynes, 2019).

**Prototypicality.** In cognitive science, prototypicality corresponds to the extent to which an entity belongs to a conceptual category (Rosch, 1973). We examine whether the dimensional value of a face is correlated with its typicality, utilizing prototypicality rated face images from the Chicago Face Database (CFD) (Ma et al., 2015; Lakshmi et al., 2021; Ma et al., 2021). Ratings correspond to the average prototypicality of a face with respect to a binary gender category (relative to others of the same race) or race category. For each category, we compute the dimensional embedding value or unnormalized attribute logit of a face from a relevant AVFS-U dimension or attribute recognition model's classification layer, respectively.

Figure 4b shows that relevant AVFS-U dimensions are positively correlated with typicality according to Spearman's $r$. Evidently, category typicality—at least for the investigated social category concepts—manifests in the AVFS dimensions from learning to predict human 3AFC similarity judgments.

**Semantic classification.** Given the evidence that AVFS-U dimensional values correlate with feature magnitude, we expect them to be useful for semantic attribute classification. To validate this hypothesis, we perform attribute prediction on the following labeled face datasets: COCO (Lin et al.,

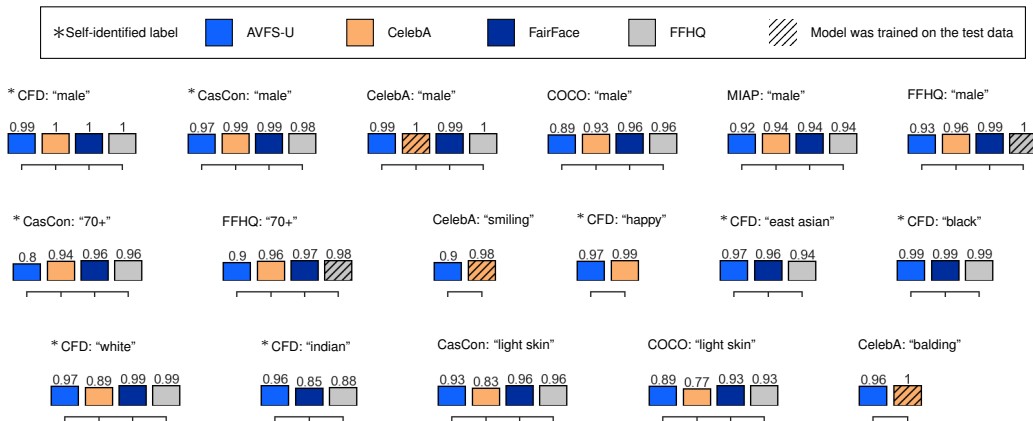

Figure 5: **Semantic classification.** AUC results are shown for binary attribute classification using continuous attribute value predictions. Our AVFS-U model is competitive with the semantically trained attribute recognition model baselines, despite not having been trained on semantic labels.

2014), OpenImages MIAP (MIAP) (Schumann et al., 2021), CFD, FFHQ, CelebA (Liu et al., 2015), and Casual Conversations (CasCon) (Hazirbas et al., 2021). Continuous attribute value predictions are computed in the same manner as in the prototypicality experiment. Results are shown in Figure 5. We observe that AVFS-U dimensions are competitive with semantically trained recognition models, even on challenging unconstrained data (e.g., COCO, MIAP).

**Attribute disparity estimation.** Next we investigate the utility of individual embedding dimensions for the task of comparing dataset attribute disparities. Suppose we are given $n$ subsets $\{\mathcal{S}_1, \ldots, \mathcal{S}_n\}$ sampled from $\mathcal{S}^*$, where, $\forall c \in [1 \ldots, n]$, $r_c \in [0, 0.01, \ldots, 0.99, 1]$ and $1 - r_c$ denote the ground-truth proportion of $\mathcal{S}_c = \{(\mathbf{x}_i, y_i)\}_i$, labeled $y = 0$ and $y = 1$, respectively. We aim to find $c$ with the lowest ground-truth attribute disparity, i.e., $\arg\min_c \Delta(c) = \arg\min_c \text{abs}(2r_c - 1)$.

We define $\text{sim}(i, j) = \text{abs}(\hat{y}_i - \hat{y}_j)^{-1}$ as the similarity between $\mathbf{x}_i$ and $\mathbf{x}_j$, where $\hat{y} \in \mathbb{R}$ is a predicted continuous attribute value for ground-truth $y \in \{0, 1\}$ given $\mathbf{x}$. $\hat{y}$ is computed in the same manner as in the prototypicality and semantic classification experiments. Let $\mathbf{K} \in \mathbb{R}^{m \times m}$ denote a similarity matrix, where $m$ denotes the number of images and $\mathbf{K}_{i,j} = \text{sim}(i, j)$. The average similarity of samples in a set $\mathcal{S}_c$, is $\propto \sum_i \sum_j \mathbf{K}_{i,j}$ (Leinster & Cobbold, 2012). We estimate $c$ as follows: $\arg\min_c \hat{\Delta}(c) = \arg\min_c |\mathcal{S}_c|^{-2} \sum_i \sum_j \mathbf{K}_{i,j}$. In all experiments, we set $n = 100$ and $m = 100$. For $\mathcal{S}^*$, we use the following labeled face datasets: COCO, MIAP, CFD, FFHQ, CelebA, and CasCon. Averaging over 100 runs, we report (1) $\Delta(\arg\min_c \hat{\Delta}(c))$; and (2) Spearman's $r$ between $\{\hat{\Delta}(1), \ldots, \hat{\Delta}(n)\}$ and $\{\Delta(1), \ldots, \Delta(n)\}$.

Results are shown in Figure 6. Despite not being trained on semantically labeled data, we observe that AVFS-U dimensions are competitive with the baselines, in particular AVFS-U disparity scores are highly correlated with ground-truth disparity scores.

## 6 DISCUSSION

Motivated by legal, technical, and ethical considerations associated with semantic labels, we introduced AVFS—a new dataset of 3AFC face similarity judgments over 4,921 faces. We demonstrated the utility of AVFS for learning a continuous, low-dimensional embedding space aligned with human perception. Our embedding space, induced under a novel conditional framework, not only enables the accurate prediction of face similarity, but also provides a human-interpretable decomposition of the dimensions used in the human-decision making process, and the importance distinct annotators place on each dimension. We further showed the practicality of the dimensions for collecting continuous attributes, performing classification, and comparing dataset attribute disparities.

Since AVFS only contains 0.003% of all possible triplets over 4,921 faces, future triplets could be actively sampled (Lindley, 1956; MacKay, 1992) so as to learn additional face-varying dimensions

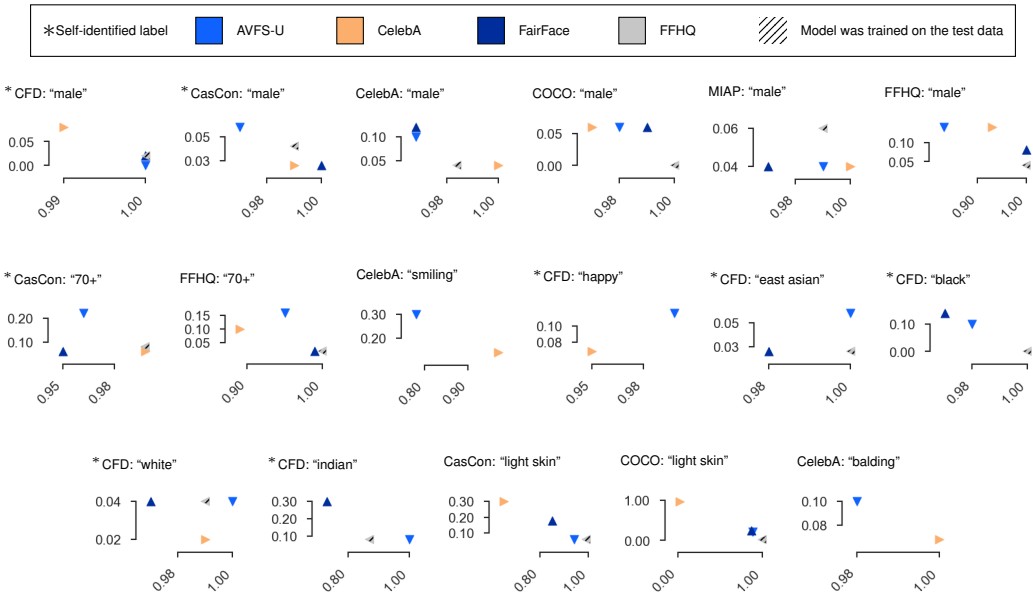

Figure 6: **Attribute disparity estimation.** Average attribute disparity ($y$-axis) in data subsets selected as the most diverse and average Spearman's $r$ ($x$-axis) between the ground-truth disparity for each subset considered and model-estimated disparity scores. Results highlight that disparity scores derived using AVFS-U model dimensions are highly correlated with ground-truth disparities.

using larger stimulus sets. For example, Roads & Love (2021) selectively sample from an approximate posterior distribution using variational inference and model ensembles. However, similar to Zheng et al. (2019)'s MDS approach, judgments are pooled, which disregards an individual annotator's decision-making process. As we have shown, annotators are (at least in our task) not interchangeable and are influenced by their sociocultural backgrounds. Future work could therefore combine our model of conditional decision-making with active sampling, e.g., seeking at each iteration unlabeled triplets that maximize expected information gain conditioned on the annotator (or annotator group) that will be assigned the labeling task.

Our work is not without its limitations. First, we implicitly assumed that the stimulus set was diverse. Therefore, our current proposal is limited to the proof of concept that we can learn face-varying dimensions that are both present within the stimuli and salient to human judgments of face similarity. Our stimulus set was sampled such that the faces were diverse with respect to inferred demographic subgroup labels. Ideally, these labels would have been self-reported. With the principle of data minimization in mind, a potential solution would be to collect stimuli from unique individuals that differ in terms of their country of residence, providing both identity- and geographic-based label diversity. Notably, country of residence information additionally helps to ensure that each image subject's data is protected appropriately (Andrews et al., 2023). Second, the number of (near) nonzero dimensions depends on the sparsity penalty weight, i.e., $\alpha_1$ in Equation (4), which must be carefully chosen. We erred on the side of caution (lower penalty) to avoid merging distinct factors, which would have reduced interpretability. Third, the emergence of a "smiling expression" and "neutral expression" dimension indicate that annotators did not always follow our instruction to ignore differences in facial expression. While this may be considered a limitation, our model was able to reveal the use of these attributes, providing a level of transparency not offered by existing face datasets. Finally, our annotator pool was demographically imbalanced, which is an unfortunate consequence of using AMT.

Despite these limitations, we hope that our work inspires others to pursue unorthodox tasks for learning the continuous dimensions of human diversity, which do not rely on problematic semantic labels. Moreover, we aim to increase discourse on annotator positionality. As the philosopher Thomas Nagel suggested, it is impossible to take a "view from *nowhere*" (Nagel, 1989). Therefore, we need to make datasets more inclusive by integrating a diverse set of perspectives from their inception.

ETHICS STATEMENT

AVFS contains annotations from a demographically imbalanced set of annotators, which may not generalize to a set of demographically dissimilar annotators. Moreover, while the stimulus set of images for which the annotations pertain to were diverse with respect to intersectional demographic subgroup labels, the labels were inferred by a combination of algorithms and human annotators. (Refer to the dataset documentation in Appendix C for details on the stimulus set and annotator distributions.) Therefore, the stimulus set is not representative of the entire human population and may in actuality be demographically biased. Although we provided evidence that models trained on AVFS generalize to novel stimuli, it is conceivable that the models could perform poorly on faces with attributes that were not salient to human judgment behavior on triplets sampled from, or represented in, the stimulus set of images. Future work could explore active learning so as to reveal additional attributes that are relevant to human judgments of face similarity and expanding the stimulus set.

Furthermore, the stimulus set (sampled from the permissively licensed FFHQ dataset) was collected without image subject consent. Reliance on such data is an overarching problem in the computer vision research community, due to a lack of consensually collected, unconstrained, and diverse human-centric image data. To preserve the privacy of the image subjects, all face images presented in this paper have been replaced by StyleGAN3 generated faces with similar embedding values.

To demonstrate that AVFS judgments encode discriminative information, we validated our learned dimensions in terms of their ability to estimate attribute disparities, as well as to predict binary attributes and concept prototypicality. Our results show that discriminative attribute classifiers can be distilled solely from learning to predict similarity judgments, underscoring that we do not need to explicitly collect face attribute labels from human annotators. Note however, for social constructs (e.g., gender, race) and emotions, we do not condone classification, as it reifies the idea that an individual's social identity or internal state can be inferred from images alone.

REPRODUCIBILITY STATEMENT

Data and code are publicly available under a Creative Commons license (CC-BY-NC-SA), permitting noncommercial use cases at `https://github.com/SonyAI/a_view_from_somewhere`.

ACKNOWLEDGMENTS

This work was funded by Sony AI.

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

# A   IMPLEMENTATION DETAILS

## A.1   FACE ALIGNMENT

For each face dataset used in this paper, faces were detected and aligned following the procedure outlined by Or-El et al. (2020), which utilizes the dlib face detector and 68 landmark shape predictor (King, 2009).

## A.2   AVFS MODELS

Aligned face images were resized to $128 \times 128$. For training, standard data augmentation was used, i.e., horizontal mirroring and $112 \times 112$ random cropping, and image values were normalized to $[-1, 1]$. Training was performed using a batch size of 128 on a single Tesla T4 GPU.

For the unconditional AVFS models, guided by the validation loss, we empirically set $\alpha_1 = 0.00005$ and $\alpha_2 = 0.01$ using grid search. The conditional models used the same hyperparameters with the additional loss term's weight set to $\alpha_3 = 0.0001$. All models were optimized for 40 epochs with Adam (Kingma & Ba, 2014) (default parameters) and learning rate 0.001.

Post-optimization, for each trained model, we obtained a core set of dimensions by removing dimensions with a maximal value (over the stimulus set) close to zero, resulting in a low-dimensional space. For the AVFS-U trained model, this resulted in 22 dimensions. The threshold for removing dimensions for all models was determined based on maximizing accuracy on the validation set. For the AVFS-U model, across five independent runs, the 95% confidence interval (CI) for the number of dimensions and validation accuracy were 27.4±5.8 and 61.9±0.1%, respectively. Figure 7 shows that the judgments can be encapsulated using a limited number of the available dimensions and that the 22 dimensions are approximately replicated across the independent runs.

Note that the AVFS-2AFC model was trained using a triplet margin with a distance swap loss (Balntas et al., 2016) for 40 epochs with Adam (Kingma & Ba, 2014) (default parameters) and learning rate 0.001.

## A.3   BASELINE MODELS

Aligned face images were resized to $128 \times 128$. For training, standard data augmentation was used, i.e., horizontal mirroring and $112 \times 112$ random cropping, and image values were normalized to $[-1, 1]$. Training was performed using a batch size of 512 across 4 Tesla T4 GPUs.

**FairFace models.** We trained face attribute recognition models using SGD for 45 epochs on the FairFace dataset. The initial learning rate 0.1 was divided by 10 at epoch 15, 30, and 40. $L^2$ weight decay and SGD momentum were set to 0.0005 and 0.9, respectively. The FairFace models were trained to predict age, gender, and ethnicity.

**CelebA models.** We trained face attribute recognition models using SGD for 45 epochs on the CelebA dataset. The initial learning rate 0.1 was divided by 10 at epoch 15, 30, and 40. $L^2$ weight

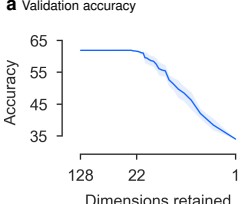
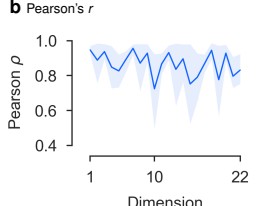
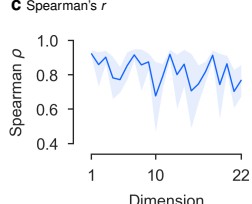

Figure 7: **AVFS-U model training across independent runs.** (a) Validation accuracy as a function of the number of dimensions retained over five independent runs (95 % CI), empirically evidencing that the judgments can be encapsulated using a limited number of dimensions. (b) Pearson and (c) Spearman's $r$ between each of the 22 dimensions and the best matching dimension from each of the other four runs (95 % CI), empirically evidencing the reproducibility of the dimensions.

decay and SGD momentum were set to 0.0005 and 0.9, respectively. The CelebA models were trained to predict 40 binary attributes.

**FFHQ models.** As FFHQ is an unlabeled dataset, we used age and gender labels inferred by human annotators from the FFHQ-Aging dataset. Ethnicity labels were inferred using the pretrained FairFace model described above. We trained the FFHQ models to predict the inferred age, gender, and ethnicity labels using SGD for 45 epochs. The initial learning rate 0.1 was divided by 10 at epoch 15, 30, and 40. $L^2$ weight decay and SGD momentum were set to 0.0005 and 0.9, respectively.

**CASIA-WebFace models.** We trained several face recognition models on the CASIA-WebFace dataset. The models differ only in terms of the loss minimized, i.e., softmax, ArcFace (Deng et al., 2019), CosFace (Wang et al., 2018), and SphereFace (Liu et al., 2017). Models were trained using SGD for 55 epochs. The initial learning rate 0.1 was divided by 10 at epoch 15, 30, and 40. $L^2$ weight decay and SGD momentum were set to 0.0005 and 0.9, respectively.

For the additional baseline models used in Appendix B, we obtained publicly available model weights for self-supervised and object recognition models. Aligned faces were preprocessed based on the model used.

**Self-supervised models.** We used the following pretrained self-supervised models:

- Self-supervised model trained using the SwAV (Caron et al., 2020) framework on ImageNet1K (Russakovsky et al., 2015)—an object recognition dataset of 1.3M images over 1K object classes, where an estimated 17% of the images contain at least one human face (Yang et al., 2022). Model weights can be found at `https://github.com/facebookresearch/swav`.
- Self-supervised models trained using the SwAV framework on IG-1B (Goyal et al., 2022)—a dataset of 1B uncurated Instagram images, containing millions of images of humans. Model weights can be found at `https://github.com/facebookresearch/vissl/tree/main/projects/SEER`.
- Self-supervised models trained using the SwAV, MoCo-v2 (Chen et al., 2020), and DINO (Caron et al., 2021) frameworks on PASS (Asano et al., 2021)—a dataset of 1.3M images without people. Model weights can be found at `https://github.com/yukimasano/PASS`.

**Object recognition models.** We used a object recognition model from PyTorch pretrained using a softmax loss on ImageNet1K (Russakovsky et al., 2015).

## B    EXTENDED EXPERIMENTAL RESULTS

For FairFace, CelebA, and FFHQ models, we additionally consider unnormalized class logit embeddings (e.g., denoted CelebA-L), as well as logit embeddings converted to binary vectors based on class predictions (e.g., denoted CelebA-BL). All baseline embeddings are normalized to unit-length, where the dot product of two embeddings determines their similarity. Results are shown in Table 1.

## C    DATASET DOCUMENTATION

The questions answered in this dataset documentation were proposed by Gebru et al. (2018).

### C.1    MOTIVATION

#### C.1.1    FOR WHAT PURPOSE WAS THE DATASET CREATED?

Few datasets contain self-identified sensitive attributes, inferring attributes risks introducing additional biases, and collecting attributes can carry legal risks. Besides, categorical labels can fail to reflect the continuous nature of human phenotypic diversity, making it difficult to compare the similarity between same-labeled faces. Motivated by these legal, technical, and ethical considerations, the dataset was created to enable research on learning human-centric face embeddings aligned with human perception, as well as studying annotator bias. The dataset does not contain problematic semantic face attribute labels, rather it consists of face similarity judgments, where each judgment

Table 1: **Predicting similarity.** (Same Stimuli) Accuracy over 24,060 judgments and Spearman's $r$ between the entropy in human- and model-generated triplet odd-one-out probabilities. (Novel Stimuli) Accuracy over 80,300 judgments and Spearman's $r$ between the entropy in human- and model-generated similarity matrices. Note that #Labels is an approximate of the number of labels collected to create a dataset accounting for consensus.

| Data/Model | Loss/Framework | #Images | #Labels | Architecture | Same Stimuli | | Novel Stimuli | |
|---|---|---|---|---|---|---|---|---|
| | | | | | Acc. | $r$ | Acc. | $r$ |
| ImageNet1K | Cross-entropy | 1.3M | >1.3M | RegNet-32Gf | 46.6 | 0.09 | 41.7 | 0.32 |
| FairFace | Cross-entropy | 62K | 372K | ResNet18 | 55.9 | 0.41 | 51.9 | 0.67 |
| FairFace-L | Cross-entropy | 62K | 372K | ResNet18 | 52.9 | 0.25 | 48.6 | 0.54 |
| FairFace-BL | Cross-entropy | 62K | 372K | ResNet18 | 51.7 | 0.15 | 47.7 | 0.54 |
| CelebA | Cross-entropy | 178K | 7.1M | ResNet18 | 52.1 | 0.25 | 48.9 | 0.56 |
| CelebA-L | Cross-entropy | 178K | 7.1M | ResNet18 | 53.2 | 0.15 | 49.6 | 0.59 |
| CelebA-BL | Cross-entropy | 178K | 7.1M | ResNet18 | 47.0 | 0.14 | 45.9 | 0.47 |
| FFHQ | Cross-entropy | 70K | 140K | ResNet18 | 50.5 | 0.16 | 47.3 | 0.49 |
| FFHQ-L | Cross-entropy | 70K | 140K | ResNet18 | 53.5 | 0.31 | 50.5 | 0.62 |
| FFHQ-BL | Cross-entropy | 70K | 140K | ResNet18 | 51.0 | 0.23 | 47.3 | 0.54 |
| CASIA-WebFace | ArcFace | 404K | 404K | ResNet18 | 51.6 | 0.29 | 46.1 | 0.40 |
| CASIA-WebFace | Cross-entropy | 404K | 404K | ResNet18 | 48.6 | 0.21 | 43.2 | 0.34 |
| CASIA-WebFace | SphereFace | 404K | 404K | ResNet18 | 48.3 | 0.25 | 43.8 | 0.35 |
| CASIA-WebFace | CosFace | 404K | 404K | ResNet18 | 44.0 | 0.12 | 40.8 | 0.23 |
| AVFS-C | Conditional Equation (4) | 5K | 574K | ResNet18 | **67.4** | **0.68** | **61.7** | 0.82 |
| AVFS-CPH | Conditional Equation (4) | 5K | 574K | ResNet18 | 66.5 | **0.68** | 61.4 | 0.82 |
| AVFS-U | Unconditional Equation (4) | 5K | 574K | ResNet18 | 62.0 | 0.65 | 57.5 | **0.86** |
| AVFS-U-50% | Unconditional Equation (4) | 5K | 287K | ResNet18 | 61.3 | 0.61 | 55.8 | 0.81 |
| AVFS-U-25% | Unconditional Equation (4) | 5K | 144K | ResNet18 | 60.6 | 0.56 | 55.3 | 0.80 |
| AVFS-U-12.5% | Unconditional Equation (4) | 5K | 72K | ResNet18 | 58.6 | 0.47 | 55.0 | 0.79 |
| AVFS-2AFC | Triplet margin with distance swap (Balntas et al., 2016) | 5K | 574K | ResNet18 | 60.2 | 0.46 | 52.8 | 0.64 |
| ImageNet1K | SwAV | 1.3M | 0 | ResNet50-w5 | 43.8 | 0.08 | 41.0 | 0.30 |
| IG-1B | SwAV | 1B | 0 | RegNet-32Gf | 47.2 | 0.18 | 44.7 | 0.45 |
| IG-1B | SwAV | 1B | 0 | RegNet-64Gf | 48.1 | 0.16 | 44.4 | 0.45 |
| IG-1B | SwAV | 1B | 0 | RegNet-128Gf | 46.8 | 0.15 | 42.8 | 0.40 |
| IG-1B | SwAV | 1B | 0 | RegNet-256Gf | 47.8 | 0.17 | 43.1 | 0.41 |
| PASS | MoCo-v2 | 1.3M | 0 | ResNet50 | 42.3 | 0.09 | 40.5 | 0.27 |
| PASS | SwAV | 1.3M | 0 | ResNet50 | 42.4 | 0.12 | 40.4 | 0.27 |
| PASS | DINO | 1.3M | 0 | ViTS-16 | 43.2 | 0.10 | 41.8 | 0.32 |

is accompanied by both the identifier and demographic attributes of the annotator who made the judgment.

### C.1.2  WHO CREATED THE DATASET AND ON BEHALF OF WHICH ENTITY?

The dataset was constructed by the authors Jerone T. A. Andrews, Przemysław Joniak, and Alice Xiang.

### C.1.3  WHO FUNDED THE CREATION OF THE DATASET?

The dataset was funded by Sony AI.

## C.2  COMPOSITION

### C.2.1  WHAT DO THE INSTANCES THAT COMPRISE THE DATASET REPRESENT?

The instances comprising the dataset represent:

- Three-alternative forced choice (3AFC) triplet-judgment tuples, where a judgment corresponds to the least similar (i.e., odd-one-out) face in a triplet of face images according to a human annotator. Refer to Figure 8 for an illustrative example of the 3AFC task presented to annotators.
- Human-generated topic labels that capture the meaning of a set of learned embedding dimensions. Refer to Figure 9 for an illustrative example of the dimension labeling task presented to annotators.
- Human-generated image ratings along a set of learned embedding dimensions. Refer to Figure 10 for an illustrative example of the image rating task presented to annotators.

All instances in the dataset are associated with the unique identifier and demographic attributes of the annotator who performed the annotation. The identifiers were randomly generated (to preserve

Choose the person that looks least similar to the two other people. Focus on the people, ignoring differences in head position, facial expression, lighting, accessories, background, and objects.

Figure 8: **Odd-one-out 3AFC task.** Annotators were tasked with choosing the person that looks least similar to the two other people. Context makes salient context-related properties and the extent to which faces being compared share these properties. By exchanging a face in a triplet (i.e., altering the context), each judgment implicitly encodes the dimensions relevant to pairwise similarity. All face images in the figure were replaced by synthetic StyleGAN3 (Karras et al., 2021) images for privacy reasons; annotators were shown real images.

Which visual characteristics do you think the people on the left-hand side of the grid have in common compared to the people on the right-hand side of the grid?

Figure 9: **Dimension labeling task.** Illustrative example of a face image grid for model dimension $j$ shown to annotators, displaying the stimulus set of 4,921 faces sorted in descending order (from left to right) based on their dimension $j$ embedding value. Annotators were tasked with writing 1–3 descriptions based on the characteristics they thought were changing along the grid. Annotators were instructed to view and interact with the entire grid before providing their descriptions. Consistent descriptions from distinct annotators would suggest that the ordering of the grid is visually meaningful. All face images in the figure were replaced by synthetic StyleGAN3 images for privacy reasons; annotators were shown real images.

the privacy of the annotators) by the authors and the demographic attributes correspond to the self-reported age, nationality, ancestry, and gender identity of an annotator.

### C.2.2 HOW MANY INSTANCES ARE THERE IN TOTAL?

The dataset contains:

- 638,180 quality-controlled 3AFC instances for training and validation; and $24,060 + 80,300$ quality-controlled 3AFC instances for testing.
- 738 quality-controlled human-generated topic labels that capture the meaning of a set of learned embedding dimensions.
- 8,800 quality-controlled human-generated image ratings along a set of learned embedding dimensions.

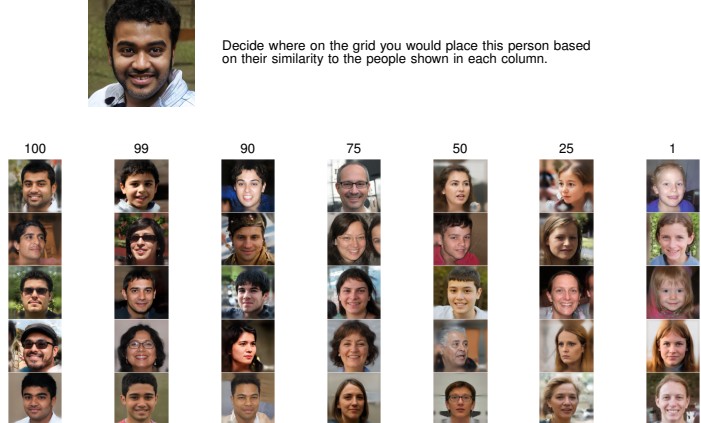

Figure 10: **Image rating task.** Illustrative example of a face image grid for model dimension $j$ shown to annotators, displaying the stimulus set of 4,921 faces sorted in descending order (from left to right) based on their dimension $j$ embedding value. Annotators were tasked with deciding which column of the grid a novel face (shown above the grid) should be placed on based on its similarity to the faces in the column. Annotators were instructed to view and interact with the entire grid before making their decision. As annotators were not primed with the meaning of a grid, accurate placement would indicate that the ordering of the grid is visually meaningful. All face images in the figure were replaced by synthetic StyleGAN3 images for privacy reasons; annotators were shown real images.

### C.2.3 DOES THE DATASET CONTAIN ALL POSSIBLE INSTANCES OR IS IT A SAMPLE OF INSTANCES FROM A LARGER SET?

**Face image stimuli.** Starting from the 70,000 FFHQ (Karras et al., 2019) images, the stimulus set of 4,921 face images was constructed by:

1. Removing faces with an absolute head yaw or pitch angle $> \pi/6$, left or right eye occlusion score $> 20$, or eyewear label. This was done to reduce the influence of head pose on human behavior and more easily permit eye comparisons.
2. Removing face images with an age label $> 19$ years old, so as to exclude images of minors.
3. Categorizing the remaining faces into 56 intersectional demographic subgroups, which were defined based on ethnicity, age, and gender labels.
4. Randomly sampling face images (without replacement) such that each subgroup would be represented in the stimulus set. This was done to ensure that the stimulus set would at least be diverse with respect to demographic subgroup labels. Moreover, face images were randomly sampled from a subgroup, as opposed to selecting the most confidently labeled, to avoid biasing the stimulus set toward stereotypical faces.

Figure 11 shows the intersectional group counts for the 4,921 face images in the stimulus set.

An additional set of 56 images and set of 20 images were sampled from FFHQ for the test set partition of 80,300 3AFC instances and 8,800 human-generated image rating instances, respectively. Both of these sets of images were sampled in the same manner as the stimulus set. In addition, they are disjoint from each other and the stimulus set of 4,921 faces. Figures 12 and 13 show the intersectional group counts for the set of 56 images and set of 20 images, respectively.

Head yaw and pitch angles, eyewear, and eye occlusion labels were obtained from the FFHQ-Aging (Or-El et al., 2020) dataset, which Or-El et al. inferred using Face++. Note that FFHQ-Aging comprises of labels for each face image contained in FFHQ. Age and gender labels were also obtained from the FFHQ-Aging dataset, which Or-El et al. inferred by employing a group of human annotators to annotate the face images. Ethnicity was estimated using a pretrained FairFace model (Appendix A).

**3AFC instances.** The training and validation set of 638,180 quality-controlled 3AFC triplet-judgment tuples contains 638,180 unique triplets, i.e., a single judgment per unique triplet. The judgments were obtained from 1,645 eligible annotators. Figure 14 shows the demographic group counts of the

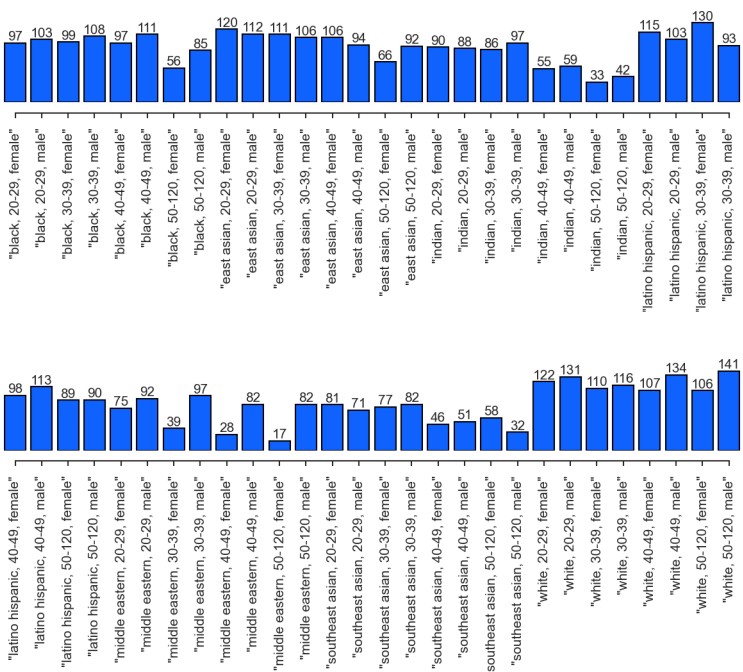

Figure 11: **Stimulus set intersectional subgroup counts.** Intersectional subgroup counts for the 4,921 face images contained in the stimulus set. There are 56 subgroups represented in the stimulus set.

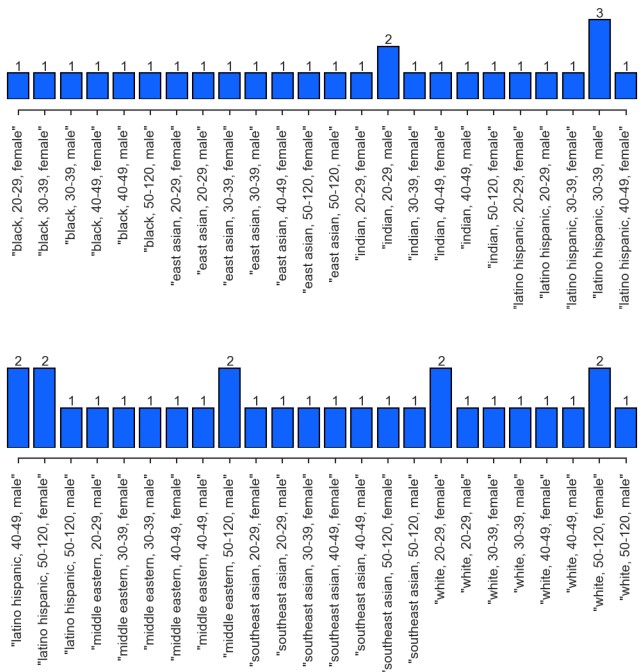

Figure 12: **Novel stimuli test set intersectional subgroup counts.** Intersectional subgroup counts for the 56 face images used in the test set partition of 80,300 quality-controlled 3AFC triplet-judgment tuples. There are 48 (from a possible 56) subgroups represented in the 27,720 unique triplets.

annotators as well as the distribution of annotator contributions.[3] The quality-controlled instances

---

[3]Note the abbreviations: "africa" ("af"), "north africa" ("afna"), "east africa" ("afea"), "middle africa" ("afma"), "south africa" ("afsa"), "western africa" ("afwa"), "americas" ("am"), "caribbean" ("amc"),"central

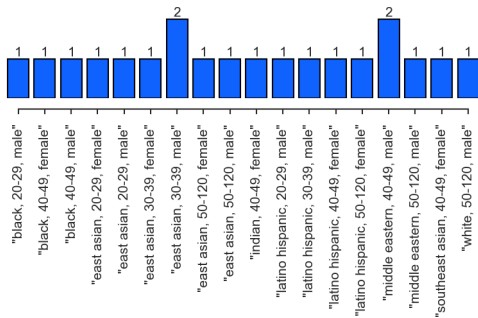

Figure 13: **Rated image set intersectional subgroup counts.** Intersectional subgroup counts for the 20 face images used for collecting human-generated image ratings along the 22 learned embedding dimensions. There are 18 (from a possible 56) subgroups represented.

Table 2: Annotator exclusion policies and criteria. 3AFC annotations from annotators who provided overly fast, deterministic, or incomplete judgments were excluded.

| Policy | Criteria | Min. #judgments |
|---|---|---|
| Fast #1 | $> 25\%$ of judgments generated in $< 0.8$s | 100 |
| Fast #2 | $> 50\%$ of judgments generated in $< 1.1$s | 100 |
| Deterministic | $> 40\%$ of judgments correspond to a single triplet position | 200 |
| Incomplete | Submission of an empty judgment | 1 |

represent a subset of instances from a larger set of 703,300 3AFC instances. The 703,300 triplets were sampled from $\binom{4,921}{3}$ possible triplets (i.e., over 4,921 face images contained in the stimulus set). Instances were not included in the subset of 638,180 3AFC instances if they were annotated by any annotator who provided overly fast, deterministic, or incomplete judgments. Refer to the annotator exclusion policies outlined in Table 2.

The test set partition of 24,060 quality-controlled 3AFC triplet-judgment tuples contains 1,000 unique triplets with 22–25 unique judgments per triplet. The judgments were obtained from 355 eligible annotators. Figure 15 shows the demographic group counts of the annotators as well as the distribution of annotator contributions. The quality-controlled instances represent a subset of instances from a larger set of 25,000 3AFC instances with 25 unique judgments per triplet. The 1,000 triplets were sampled from $\binom{4,921}{3}$ possible triplets (i.e., over 4,921 face images contained in the stimulus set). Within the 1,000 triplets, 2,163 face images from the stimulus set of 4,921 face images were used. The 1,000 triplets are disjoint from the 703,300 triplets sampled for the training and validation set. Instances were not included in the subset of 24,060 3AFC instances if they were annotated by any annotator who provided overly fast, deterministic, or incomplete judgments. Refer to the annotator exclusion policies outlined in Table 2.

The test set partition of 80,300 quality-controlled 3AFC triplet-judgment tuples contains $\binom{56}{3} = 27,720$ unique triplets, representing all possible triplets that could be sampled from 56 images. Each triplet has 2–3 judgments from 632 eligible annotators. Figure 16 shows the demographic group counts of the annotators as well as the distribution of annotator contributions. The quality-controlled instances represent a subset of instances from a larger set of 83,160 3AFC instances with 3 unique judgments per triplet. Instances were not included in the subset of 80,300 3AFC instances if they were annotated by any annotator who provided overly fast, deterministic, or incomplete judgments. Refer to the annotator exclusion policies outlined in Table 2.

**Dimension topic label instances.** The 738 quality-controlled human-generated topic labels were obtained for 22 learned embedding dimensions, representing a subset from 128 dimensions. Figure 17 shows the learned embeddings with the highest frequency topic labels generated by the annotators.

america" ("amca"), "south america" ("amasa"), "northern america" ("amna"), "asia" ("as"), "central asia" ("asca"), "eastern asia" ("asea"), "south-eastern asia" ("assea"), "southern asia" ("assa"), "western asia" ("aswa"), "europe" ("eu"), "eastern europe" ("euea"), "northern europe" ("eune"), "southern europe" ("euse"), "western europe" ("euwe"), "oceania" ("oc"), "australia and new zealand" ("ocanz"), "melanesia" ("ocme"), "micronesia" ("ocmi"), and "polynesia" ("ocp").

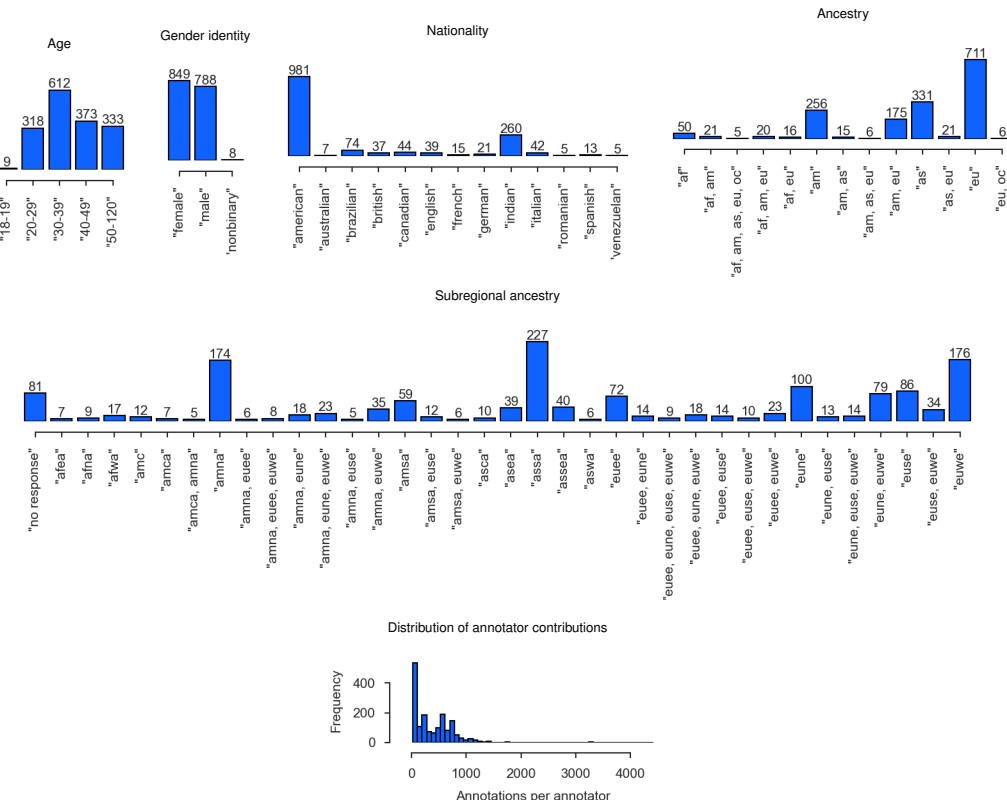

Figure 14: **3AFC training and validation set annotator demographics.** Annotator age, gender identity, nationality, ancestry, and subregional ancestry group counts for the 1,645 eligible annotators who contributed to the training and validation set of 638,180 quality-controlled 3AFC triplet-judgment tuples. The distribution of per annotator contributions is also shown. Note that for nationality, ancestry, and subregional ancestry the plots only show groups with $\geq 5$ annotators.

The subset of dimensions were selected on the basis of their maximal observed value (over the stimulus set of 4,921 faces) being sufficiently larger than zero. The topic labels were obtained from 84 eligible annotators. Figure 18 shows the demographic group counts of the annotators as well as the distribution of annotator contributions. Each of the 22 dimensions has 23–58 topic labels. The quality-controlled topic labels were arrived at by first removing annotator responses that did not correspond to words and those that were derogatory or offensive. The filtered responses were then converted into 35 topic labels: "age", "age related", "ancestry", "ear shape", "emotion expression", "eye color", "eye related", "eye shape", "eyebrow related", "eyebrow shape", "face related", "face shape", "facial expression", "facial hair", "forehead shape", "gender expression", "hair color", "hair length", "hair related", "hair texture", "hairstyle", "head shape", "health related", "lips related", "lips shape", "mouth related", "mouth shape", "neck related", "nose related", "nose shape", "skin color", "skin related", "skin texture", "teeth related", and "weight related".

**Dimension rating labels.** The 8,800 quality-controlled human-generated image ratings were obtained for 22 learned embedding dimensions, representing a subset from 128 dimensions. The subset of dimensions were selected on the basis of their maximal observed value (over the stimulus set of 4,921 faces) being sufficiently larger than zero. The ratings were obtained from 164 eligible annotators. Figure 19 shows the demographic group counts of the annotators as well as the distribution of annotator contributions. Quality control corresponded to excluding incomplete responses, i.e., empty responses. The 8,800 ratings represent all of the ratings collected from annotators, i.e., all collected ratings were nonempty.

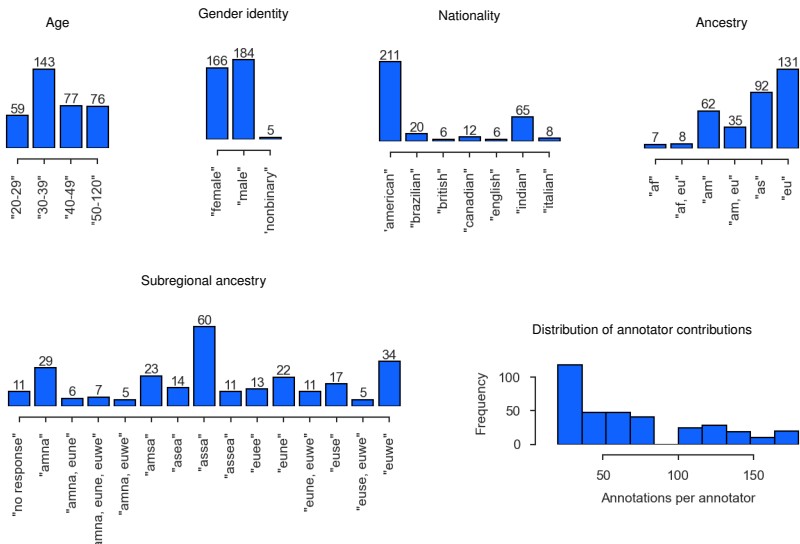

Figure 15: **3AFC same stimuli test set annotator demographics.** Annotator age, gender identity, nationality, ancestry, and subregional ancestry group counts for the 355 eligible annotators who contributed to the test set partition of 24,060 quality-controlled 3AFC triplet-judgment tuples. The distribution of per annotator contributions is also shown. Note that for nationality, ancestry, and subregional ancestry the plots only show groups with $\geq 5$ annotators.

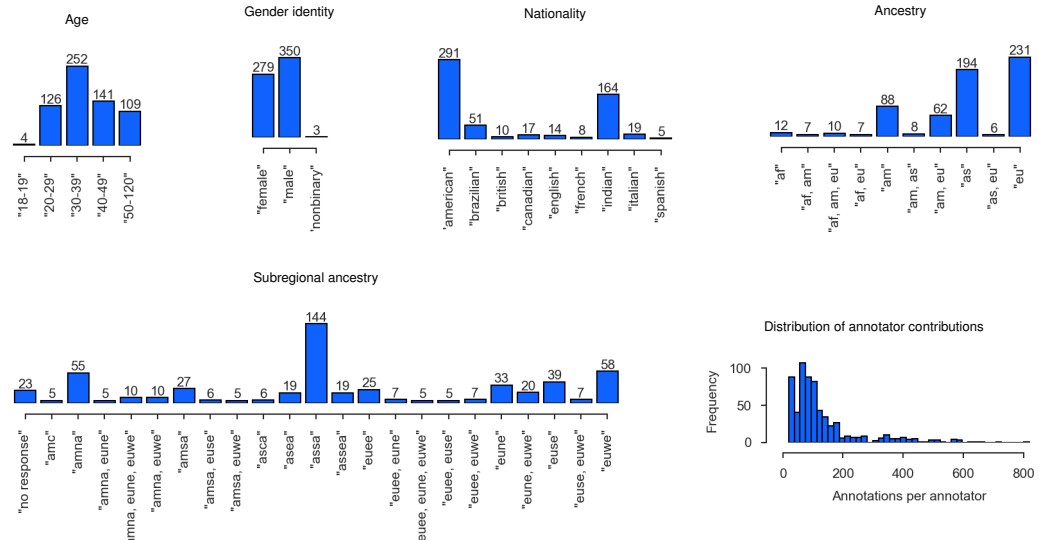

Figure 16: **3AFC novel stimuli test set annotator demographics.** Annotator age, gender identity, nationality, ancestry, and subregional ancestry group counts for the 632 eligible annotators who contributed to the test set partition of 80,300 quality-controlled 3AFC triplet-judgment tuples. The distribution of per annotator contributions is also shown. Note that for nationality, ancestry, and subregional ancestry the plots only show groups with $\geq 5$ annotators.

### C.2.4 WHAT DATA DOES EACH INSTANCE CONSIST OF?

- Each 3AFC instance consists of a triplet of face image identifiers, 3AFC (i.e., odd-one-out) judgment, and annotator identifier (corresponding to the annotator who generated the topic label).

- Each human-generated topic label instance consists of a dimension identifier, topic label, and annotator identifier (corresponding to the annotator who generated the topic label).

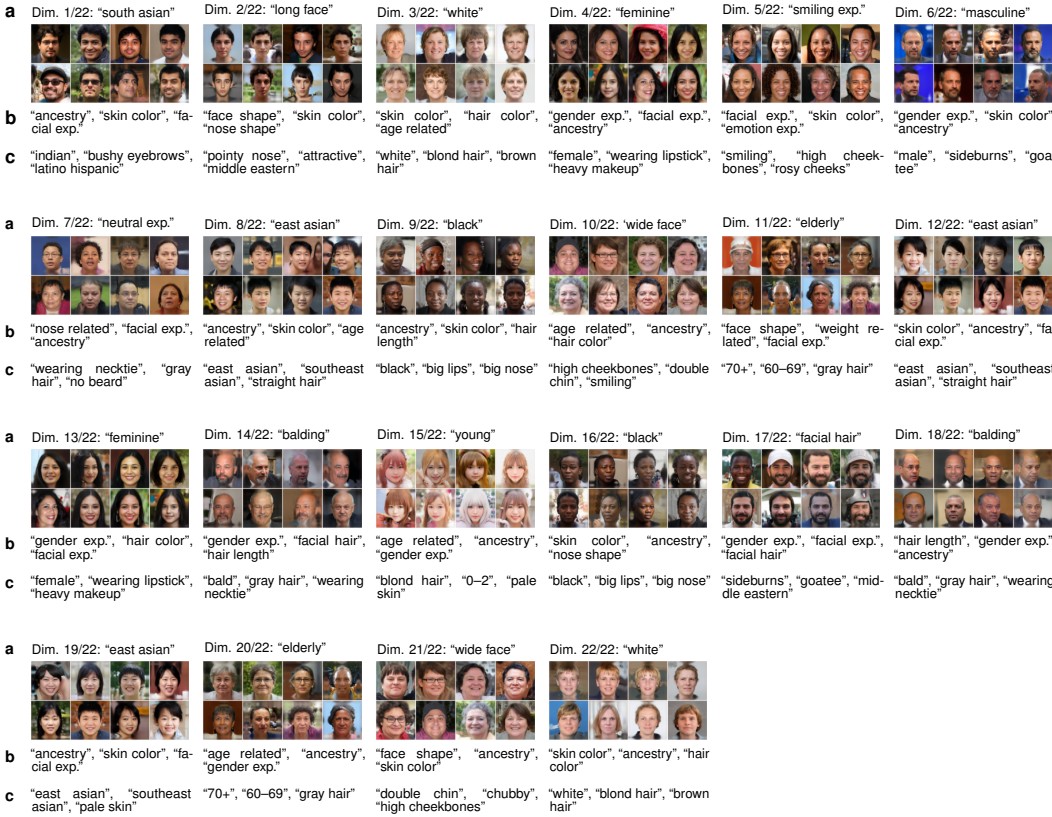

Figure 17: **Dimensional interpretability.** (a) Our interpretation of an AVFS-U dimension with the highest scoring stimulus set images. (b) Highest frequency human-generated topic labels. (c) CelebA and FairFace model-generated labels, exhibiting a clear correspondence with human-generated topics. Face images in this figure have been replaced by synthetic StyleGAN3 images for privacy reasons.

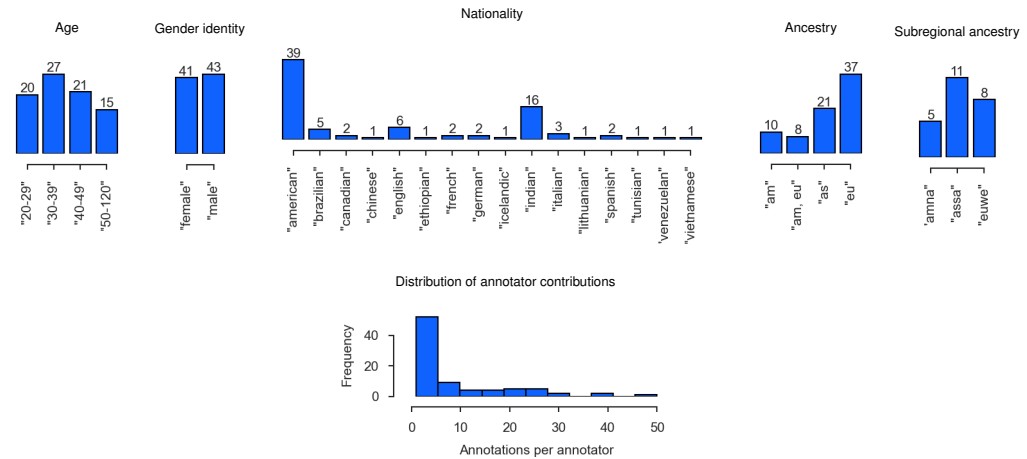

Figure 18: **Topic labeling annotator demographics.** Annotator age, gender identity, nationality, ancestry, and subregional ancestry group counts for the 84 eligible annotators who contributed 738 human-generated topic labels that capture the meaning of the 22 learned embedding dimensions. The distribution of per annotator contributions is also shown. Note that for nationality, ancestry, and subregional ancestry the plots only show groups with $\geq 5$ annotators.

- Each human-generated image rating instance consists of a dimension identifier, image identifier, image rating, and annotator identifier (corresponding to the annotator who generated the topic label).

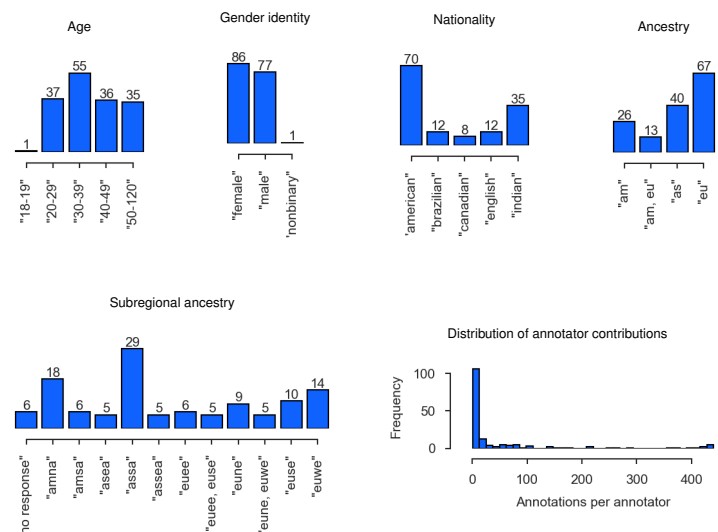

Figure 19: **Image rating annotator demographics.** Annotator age, gender identity, nationality, ancestry, and subregional ancestry group counts for the 164 eligible annotators who contributed 8,800 human-generated image ratings along the 22 learned embedding dimensions. The distribution of per annotator contributions is also shown. Note that for nationality, ancestry, and subregional ancestry the plots only show groups with $\geq 5$ annotators.

Note that each annotator's identifier is associated with their demographic attributes (i.e., age, gender identity, nationality, and ancestry).

### C.2.5 IS THERE A LABEL OR TARGET ASSOCIATED WITH EACH INSTANCE?

- Each 3AFC instance's target is a 3AFC judgment.
- Each human-generated topic label instance's target is a topic label.
- Each human-generated image rating instance's target is an image rating.

### C.2.6 IS ANY INFORMATION MISSING FROM INDIVIDUAL INSTANCES?

No.

### C.2.7 ARE RELATIONSHIPS BETWEEN INDIVIDUAL INSTANCES MADE EXPLICIT?

Yes. Each instance can be mapped to the annotator who labeled it.

### C.2.8 ARE THERE RECOMMENDED DATA SPLITS?

The 3AFC instances are split into training, validation, and testing partitions. All other instances are for evaluating learned model dimensions, i.e., for the model introduced in the paper.

### C.2.9 ARE THERE ANY ERRORS, SOURCES OF NOISE, OR REDUNDANCIES IN THE DATASET?

No.

### C.2.10 IS THE DATASET SELF-CONTAINED, OR DOES IT LINK TO OR OTHERWISE RELY ON EXTERNAL RESOURCES?

The dataset does not include any images, it instead references to image identifiers. The image identifiers can be used to obtain the relevant FFHQ images, which are hosted on NVIDIA Corporation's Google Drive: https://github.com/NVlabs/ffhq-dataset.

### C.2.11 Does the dataset contain data that might be considered confidential?

The dataset includes the demographic information (i.e., age, nationality, ancestry, and gender identity) of each annotator; however, all annotators consented to the collection, use, and publication of their data.

### C.2.12 Does the dataset contain data that, if viewed directly, might be offensive, insulting, threatening, or might otherwise cause anxiety?

No.

### C.2.13 Does the dataset relate to people?

The dataset contains annotations corresponding to human judgments of face similarity, as well as demographic information about each annotator.

### C.2.14 Does the dataset identify any subpopulations?

The dataset includes the demographic information (i.e., age, nationality, ancestry, and gender identity) of each annotator.

### C.2.15 Is it possible to identify individuals, either directly or indirectly from the dataset?

Each annotator's Amazon Mechnical Turk (AMT) identifier was replaced by an identifier randomly generated by the authors. The dataset therefore contains de-identified data.

### C.2.16 Does the dataset contain data that might be considered sensitive in any way?

The dataset includes the demographic information (i.e., age, nationality, ancestry, and gender identity) of each annotator.

## C.3 Collection process

### C.3.1 How was the data associated with each instance acquired?

All data associated with each instance was acquired via AMT, except for the image identifiers which were obtained from the FFHQ dataset: https://github.com/NVlabs/ffhq-dataset.

### C.3.2 What mechanisms or procedures were used to collect the data?

The AMT API was used to collect the data associated with each instance.

### C.3.3 If the dataset is a sample from a larger set, what was the sampling strategy?

Refer to Appendix C.2.3.

### C.3.4 Who was involved in the data collection process and how were they compensated?

The authors collected the annotations directly from annotators via AMT. All annotators were required to have previously completed $\geq 100$ human intelligence tasks (HITs) on AMT with $\geq 95\%$ approval rating. Eligibility was further determined through a prescreening survey, which included a short multiple-choice English language proficiency test. Since we placed no restriction on annotator location, in order to be eligible, we required annotators to answer at least two out of three multiple-choice English language proficiency questions correctly.

Annotators were paid a nominal fee of 0.01 USD for completing a prescreening survey. For all subsequent tasks, annotators were compensated at a rate of 15 USD per hour, regardless of whether their work passed quality control checks.

### C.3.5 OVER WHAT TIMEFRAME WAS THE DATA COLLECTED?

From 6 November 2021 to 20 February 2022.

### C.3.6 WERE ANY ETHICAL REVIEW PROCESSES CONDUCTED?

No.

### C.3.7 DID YOU COLLECT THE DATA FROM THE INDIVIDUALS IN QUESTION DIRECTLY, OR OBTAIN IT VIA THIRD PARTIES OR OTHER SOURCES?

All data was collected directly from the annotators via AMT.

### C.3.8 WERE THE INDIVIDUALS IN QUESTION NOTIFIED ABOUT THE DATA COLLECTION?

Yes. Annotators were provided with (1) an information sheet and consent form; and (2) a notification and consent for collection and use of study data form.

### C.3.9 DID THE INDIVIDUALS IN QUESTION CONSENT TO THE COLLECTION AND USE OF THEIR DATA?

Annotators consented to the collection, use, and publication of their age, gender identity, nationality, ancestry, and task responses.

### C.3.10 IF CONSENT WAS OBTAINED, WERE THE CONSENTING INDIVIDUALS PROVIDED WITH A MECHANISM TO REVOKE THEIR CONSENT IN THE FUTURE OR FOR CERTAIN USES?

Annotators were instructed to contact Sony Europe B.V. at Taurusavenue 16, 2132LS Hoofddorp, Netherlands or `privacyoffice.SEU@sony.com` to revoke their consent in the future or for certain uses.

### C.3.11 HAS AN ANALYSIS OF THE POTENTIAL IMPACT OF THE DATASET AND ITS USE ON DATA SUBJECTS BEEN CONDUCTED?

A data privacy impact assessment was conducted.

### C.4 PREPROCESSING/CLEANING/LABELING

### C.4.1 WAS ANY PREPROCESSING/CLEANING/LABELING OF THE DATA DONE?

To control for quality, any annotator who provided overly fast, deterministic, or incomplete responses were not included in the dataset. In addition, labels for model dimensions collected from human annotators were cleaned and converted to topic labels.

### C.4.2 WAS THE "RAW" DATA SAVED IN ADDITION TO THE PREPROCESSED/CLEANED/LABELED DATA?

The authors retain a copy of the raw data.

### C.4.3 IS THE SOFTWARE THAT WAS USED TO PREPROCESS/CLEAN/LABEL THE DATA AVAILABLE?

No.

## C.5 USES

### C.5.1 HAS THE DATASET BEEN USED FOR ANY TASKS ALREADY?

In the paper, the dataset was used to concurrently learn a face embedding space aligned with human perception as well as annotator-specific subspaces. The combination of the learned embedding space and annotator-specific subspaces not only enabled the accurate prediction of face similarity, but also provided a human-interpretable decomposition of the dimensions used in the human decision-making process, as well as the importance distinct annotators placed on each dimension. The paper demonstrated that the individual embedding dimensions (1) are related to concepts of gender, ethnicity, age, as well as face and hair morphology; and (2) can be used to collect continuous attributes, perform classification, and compare dataset attribute disparities. The paper further showed that annotators are influenced by their sociocultural backgrounds, underscoring the need for diverse annotator groups to mitigate bias.

### C.5.2 IS THERE A REPOSITORY THAT LINKS TO ANY OR ALL PAPERS OR SYSTEMS THAT USE THE DATASET?

Refer to https://github.com/SonyAI/a_view_from_somewhere.

### C.5.3 WHAT (OTHER) TASKS COULD THE DATASET BE USED FOR?

The dataset could be used for any task related to human perception, human decision-making processes, metric learning, active learning, bias mitigation, or disentangled representations.

### C.5.4 IS THERE ANYTHING ABOUT THE COMPOSITION OF THE DATASET OR THE WAY IT WAS COLLECTED AND PREPROCESSED/CLEANED/LABELED THAT MIGHT IMPACT FUTURE USES?

The dataset contains annotations for images contained in the FFHQ dataset of Flickr images. The images currently have permissive licenses that allow free use, redistribution and adaptation for noncommercial purposes, however Flickr users are free to request the removal of any of their images contained in the FFHQ dataset by visiting https://github.com/NVlabs/ffhq-dataset.

Moreover, the dataset contains annotations from a demographically imbalanced set of annotators, which may not generalize to a set of demographically dissimilar annotators. Furthermore, while the stimulus set of images for which the annotations pertain to were diverse with respect to intersectional demographic subgroup labels, the labels were inferred by a combination of algorithms and human annotators. Therefore, the stimulus set is not representative of the entire human population and may in actuality be demographically biased.

### C.5.5 ARE THERE TASKS FOR WHICH THE DATASET SHOULD NOT BE USED?

The dataset was created for noncommercial research purposes only. The dataset is not intended for, and should not be used for, development or improvement of facial recognition technologies.

## C.6 DISTRIBUTION

### C.6.1 WILL THE DATASET BE DISTRIBUTED TO THIRD PARTIES OUTSIDE OF THE ENTITY ON BEHALF OF WHICH THE DATASET WAS CREATED?

The dataset can be accessed by visiting https://github.com/SonyAI/a_view_from_somewhere.

### C.6.2 HOW WILL THE DATASET BE DISTRIBUTED?

The dataset can be accessed by visiting https://github.com/SonyAI/a_view_from_somewhere.

### C.6.3 WHEN WILL THE DATASET BE DISTRIBUTED?

May 2023.

### C.6.4 WILL THE DATASET BE DISTRIBUTED UNDER A COPYRIGHT OR OTHER INTELLECTUAL PROPERTY (IP) LICENSE, AND/OR UNDER APPLICABLE TERMS OF USE (ToU)?

The dataset is distributed under a Creative Commons BY-NC-SA license: https://creativecommons.org/licenses/by-nc-sa/4.0/.

### C.6.5 HAVE ANY THIRD PARTIES IMPOSED IP-BASED OR OTHER RESTRICTIONS ON THE DATA ASSOCIATED WITH THE INSTANCES?

No.

### C.6.6 DO ANY EXPORT CONTROLS OR OTHER REGULATORY RESTRICTIONS APPLY TO THE DATASET OR TO INDIVIDUAL INSTANCES?

The authors are not aware of any controls or restrictions.

## C.7 MAINTENANCE

### C.7.1 WHO WILL BE SUPPORTING/HOSTING/MAINTAINING THE DATASET?

The dataset is supported by the authors and can be accessed by visiting https://github.com/SonyAI/a_view_from_somewhere.

### C.7.2 HOW CAN THE CURATOR OF THE DATASET BE CONTACTED (E.G., EMAIL ADDRESS)?

Correspondence to jerone.andrews@sony.com.

### C.7.3 IS THERE AN ERRATUM?

If errors are found, an erratum file will be added to the dataset.

### C.7.4 WILL THE DATASET BE UPDATED?

The dataset will be versioned.

### C.7.5 IF THE DATASET RELATES TO PEOPLE, ARE THERE APPLICABLE LIMITS ON THE RETENTION OF THE DATA ASSOCIATED WITH THE INSTANCES?

No. However, annotators are free to withdraw their consent at any time and for any reason.

### C.7.6 WILL OLDER VERSIONS OF THE DATASET CONTINUE TO BE SUPPORTED/HOSTED/MAINTAINED?

Older versions of the dataset will continue to be hosted, unless they contain information about, or obtained from, an annotator who has subsequently withdrawn their consent.

### C.7.7 IF OTHERS WANT TO EXTEND/AUGMENT/BUILD ON/CONTRIBUTE TO THE DATASET, IS THERE A MECHANISM FOR THEM TO DO SO?

Others should contact the authors if they wish to contribute to the dataset.

