# OpenReview forum: "A View From Somewhere: Human-Centric Face Representations"
_ICLR.cc/2023/Conference — ICLR 2023 poster_

### Official Review · Reviewer_r72x · 2022-10-22

**Confidence:** 3
**Clarity, Quality, Novelty And Reproducibility:** The paper’s presentation is clear. It…
**Correctness:** 3
**Technical Novelty And Significance:** 3
**Empirical Novelty And Significance:** 3
**Recommendation:** 8

**Strength And Weaknesses:**

Strength:

The presentation of this paper is clear. The design of the dataset and the experiment is introduced in great details.

The authors have provided detailed supplementary materials to support the main paper.


Weakness:

It is not very clear to me how the proposed method is used for auditing dataset diversity, and how well/effective such audition will be, in terms of boosting the performance of related computer models. I wonder the correlation between human perceptual face similarity and objective machine classifications.

The previous studies on face similarity should be mentioned and discussed, for example, the following papers:


Somai, Rosyl S., and Peter JB Hancock. "Exploring perceived face similarity and its relation to image-based spaces: an effect of familiarity." Journal of Vision 21.9 (2021): 2149-2149.

Sadovnik, Amir, et al. "Finding your lookalike: Measuring face similarity rather than face identity." Proceedings of the IEEE Conference on Computer Vision and Pattern Recognition Workshops. 2018.

The authors are honest in pointing out several limitations. The efficacy of this work is based on the hypothesis that the proposed stimuli set is sufficiently diverse and there is no ideal way to prove this. Perhaps the authors can tune down their claim a bit.

 It is not clear to me why the authors chose the “odd-one-out” in triple group paradigm, though the authors have mentioned earlier work using the similar method. Why not use ranking, or “odd-one-out” in a quadruple group?


**Summary Of The Paper:**

In this paper, the authors proposed a method for implicitly learning continuous face-varying dimensions, without the need of asking an annotator to explicitly categorize a person. The result can be used in auditing datasets for diversity. The authors also proposed FAX, a novel dataset of 638,180 face similarity judgments over 4,921 faces. This dataset can be of interest for researchers from multiple disciplines.

**Summary Of The Review:**

The paper has good contributions, though its actual efficacy in real application is not very clear.

---

> ### Author Response · Authors · 2022-11-12
> **Response [1/3]**
>
> Thanks a lot for your time and effort in reviewing our paper. We appreciate it. In addition, we are pleased that you appreciate the clarity of the paper, the effort put into the design of our dataset, and the level of detail provided, which you have reflected by your current recommendation.
>
> Below are responses to the concerns raised by you. Please let us know if you require any further information, or if anything is unclear.
>
> > The previous studies on face similarity should be mentioned and discussed
>
> **Thank you for pointing out the two papers, we have added mentions to both in our related work section (lines 92-93 in our updated paper)**. Please let us know if this is satisfactory to you, if not we are happy to expand the discussion either in the related work section or appendix.
>
> **Having reviewed both papers, we made the following observations):**
>
>
> **Somai’s** work focuses on synthetic stimuli in a face identity recognition setting, i.e., “are these two faces the same person?” where one face has a PCA or ICA transformation applied. The PCA space was calculated from 72 eye-aligned images, each an average of 10 images of a famous (mostly Caucasian) female. Notably, our dimensions can be thought of as the principal components of faces that are important for human judgments of face similarity, i.e., since they explain the variation in human judgments and are predictive of the judgments. Therefore, it is possible to use our dimensions, e.g., to find latent directions in a generative model to alter the attributes of a face. This has been done using the classifier outputs from models such as CelebA. Since our dimensions individually act like classifiers, we can also do the same.
>
>
> Unlike our dataset, **Sadovnik’s dataset** is not publicly available and only contains 5,000 similarity judgments. Therefore, our dataset will be the first publicly available face similarity dataset and is orders of magnitude larger than Sadovnik’s private dataset. In addition, Sadovnik also focuses on a face recognition setting, i.e., they only collect judgments for faces that are close to each in a pretrained face recognition model’s embedding space. Another noticeable difference is that Sadovnik’s collect judgments for a ranking task. One issue with their ranking task is that it is a harder task than the odd-one-out task we use. In particular, the odd-one-out task requires an annotator to make three pairwise similarity judgments (i.e., (sim(i,j), sim(i,k), and sim(j,k)). However, the ranking task which has a query image and six images to be ranked requires six pairwise similarity judgments, i.e., between each image to be ranked and the query. Asking an annotator to maintain six pairwise similarity scores in their head is more difficult than maintaining three pairwise similarity scores. In addition, our task only necessitates an annotator to select a single image. However, Sadovnik’s ranking task requires an annotator to select and order six images, thus increasing the annotation time.
>
>
> What is learned from the ranking task may also be brought into question. Let q be the query image and the images to be ranked are $r_1, r_2, r_3, r_4, r_5, and r_6$. W.l.o.g., let’s say an annotator determines that $r_1$ is closest to $q$. Afterwards, the annotator could just compare $r_1$ to $r_2, r_3, r_4, r_5$, and $r_6$ instead of comparing $q$ to $r_2, r_3, r_4, r_5$, and $r_6$. Sadovnik could have instead designed their task differently to ensure that annotators do not follow this strategy. For instance, once the first image was selected as being closest to $q$, the image should have been removed such that the annotators could then focus on which of $r_2, r_3, r_4, r_5$, and $r_6$ are closest to $q$. This is particularly problematic as Sadovnik transforms the collected ranking judgments into triplets with anchors, where the anchor corresponds to a query image. However, it is not apparent whether the negative example in the triplet is actually a negative, since the negative may have never been compared to the query. That is, the negative may have been compared to one of the other images that were being ranked.

---

> > ### Author Response · Authors · 2022-11-12
> > **Response [2/3]**
> >
> > >It is not clear to me why the authors chose the “odd-one-out” in triple group paradigm, though the authors have mentioned earlier work using the similar method. Why not use ranking, or “odd-one-out” in a quadruple group?
> >
> >
> > Human perception of similarity can vary with respect to context [1,2]. For example, [2] found that a snake and raccoon were judged to be less similar when participants were not given any context. However, when the similarity comparison was contextualized, i.e., how similar are snakes and raccoons in the context of “pets” their similarity increased. The odd-one-out similarity task also provides context, for instance, it requires annotators to perform pairwise comparisons in order to determine the odd-one-out. By varying the context, e.g., face C in the triplet (A, B, C), we uncover different sets of features that contribute to the similarity and dissimilarity between A and B. This is important as there are an uncountable number of ways in which two objects may be similar or dissimilar [3]. Odd-one-out similarity judgments [4,5,6,7], or more generally contextual similarity comparisons, can be used to uncover the salient features that contribute to pairwise similarity. As highlighted in several works [1,8,9,10], context makes salient: (1) context-related properties; and (2) the extent to which objects being compared share context-related properties.
> >
> > The odd-one-out in a quadruple group task and a ranking task are also contextual similarity tasks. Therefore, they could be used instead. The odd-one-out in a quadruple requires an annotator to determine the three faces that are most similar. This increases the difficulty of the task, especially when all four faces are perceptually different. This task requires more time for an annotator to arrive at a judgment and the signal learned is likely to be noisier. In particular, as an annotator could think that one person is the most different from 2 of the people for one reason and the other person for a different reason. Finally, the quadruple task does not tell us about the pairwise similarity of faces. For instance, if a face is selected as the odd-one-out then we know the other three faces can be considered similar. However, we do not know which of the three faces are most similar. Of course the judgments will still encode information about how humans determine similarity, but the signal would be noisier as we cannot infer the most similar data points from the judgment. Furthermore, if we were to collect quadruple odd-one-out judgments over 4,921 images, there would be 24,404,649,407,810 possible quadruples. By using triplet odd-one-out judgments, the search space is orders of magnitude smaller, i.e., 19,849,247,180 possible triplets.
> >
> >
> > We highlighted the possible issues with the ranking task above, i.e., the ranking task used by Sadovnik.
> >
> > _____________________________________________________________________________________
> >
> > [1] Barsalou, Lawrence. "The instability of graded structure: Implications for the nature of concepts." (1987).\
> > [2] Roth, Emilie M., and Edward J. Shoben. "The effect of context on the structure of categories." Cognitive psychology 15.3 (1983): 346-378.\
> > [3] Love, Bradley C., and Brett D. Roads. "Similarity as a Window on the Dimensions of Object Representation." Trends in Cognitive Sciences 25.2 (2021): 94-96.\
> > [4] Josephs, Emilie L., Martin N. Hebart, and Talia Konkle. "Emergent dimensions underlying human perception of the reachable world." Journal of Vision 21.9 (2021): 2154-2154.\
> > [5] Hebart, Martin N., et al. "Revealing the multidimensional mental representations of natural objects underlying human similarity judgements." Nature human behaviour 4.11 (2020): 1173-1185.\
> > [6] Zheng, Charles Y., et al. "Revealing interpretable object representations from human behavior." International Conference on Learning Representations. 2018.\
> > [7] Dima, Diana C., Martin N. Hebart, and Leyla Isik. "A data-driven investigation of human action representations." bioRxiv (2022).\
> > [8] Markman, Arthur B., and Dedre Gentner. "Nonintentional similarity processing." The new unconscious (2005): 107-137.\
> > [9] Goodman, Nelson. "Seven strictures on similarity." (1972).\
> > [10] Medin, Douglas L., Robert L. Goldstone, and Dedre Gentner. "Respects for similarity." Psychological review 100.2 (1993): 254.

---

> > > ### Author Response · Authors · 2022-11-12
> > > **Response [3/3]**
> > >
> > > >The authors are honest in pointing out several limitations. The efficacy of this work is based on the hypothesis that the proposed stimuli set is sufficiently diverse and there is no ideal way to prove this. Perhaps the authors can tune down their claim a bit.
> > >
> > >
> > > We absolutely agree that there is no ideal way to prove this. **We have updated Section 4. In particular, we state the following (see lines 386-389 in the updated paper):**
> > > Our current proposal is limited to the proof of concept that our approach can only uncover factors that vary in the data and are salient to human perception of face similarity. For instance, if skin color does not vary among data instances, then skin color cannot possibly influence human judgments. Therefore, it would be impossible to learn skin color as a face-varying dimension.
> > >
> > >
> > >
> > > >It is not very clear to me how the proposed method is used for auditing dataset diversity, and how well/effective such audition will be, in terms of boosting the performance of related computer models. I wonder the correlation between human perceptual face similarity and objective machine classifications.
> > >
> > > If possible, please could you clarify what you mean by the correlation between human perceptual face similarity and objective machine classifications. In particular, what do you mean by objective machine classifications? Thank you. Once this is clarified, we will be happy to answer your question!
> > >
> > >
> > > >The paper has good contributions, though its actual efficacy in real application is not very clear.
> > >
> > > If the reviewer has any suggestions on what they would like to see, please feel free to let us know.

---

### Official Review · Reviewer_pZsn · 2022-10-25

**Confidence:** 3
**Correctness:** 4
**Technical Novelty And Significance:** 4
**Empirical Novelty And Significance:** 4
**Recommendation:** 6

**Clarity, Quality, Novelty And Reproducibility:**

Writing is clear. Novelty is good. Not sure whether the author is going to release the proposed dataset or code.

**Strength And Weaknesses:**

Strength:
The proposed topic of human-centric face representation is interesting, and the dataset introduced in this paper will contribute to the community.
Weakness:
In this paper, the proposed embedding is based on a weak human annotation (choose the person that looks least similar to the two other people, a.k.a, odd-one-out). How is the proposed embedding different from the self-supervised learning-based face embedding?


**Summary Of The Paper:**

This paper introduces a way to implicitly learn a set of continuous face-varying dimension with weak human annotations. By introducing a novel large-scale novel dataset, authors have shown the learned implicit face embeddings are human-interpretable and related to gender, race, age, as well as face and hair morphology categories.

**Summary Of The Review:**

This paper introduces a way to learn human-centric face representations. By introducing a novel dataset with weak human annotation, the learned embeddings are proved to be human-interpretable and related to gender, race, age, as well as face and hair morphology categories. The proposed topic of will contribute to the evaluation of data diversity and the dataset introduced in this paper will contribute to the community. Therefore, I recommend this paper to be accepted.

---

> ### Author Response · Authors · 2022-11-12
> **Response [1/2]**
>
> Thank you for reviewing our paper, we appreciate it! Thank you for recommending that our paper be accepted. We are happy that you think that our approach to learning face representations is interesting and that our dataset has the ability to contribute to the machine learning community.
>
> Below are responses to both concerns/questions raised by you. Please let us know if you require any further information, or if anything is unclear.
>
> >Weakness: In this paper, the proposed embedding is based on a weak human annotation (choose the person that looks least similar to the two other people, a.k.a, odd-one-out). How is the proposed embedding different from the self-supervised learning-based face embedding?
>
> Table 14 in the Appendix contains results from several baseline self-supervised methods. Self-supervised approaches correspond to MoCo-v2, SwAV, or DINO. For data, self-supervised approaches were trained on IG-1B (https://arxiv.org/abs/2202.08360), ImageNet, or PASS (https://arxiv.org/abs/2109.13228). We used the official publicly available pretrained models.
>
> SwAV models trained on IG-1B obtain the best performance among the self-supervised approaches. IG-1B has 1 billion images and contains at least tens of millions of images of people that are apparently from diverse places with a range of gender expressions. (A random subsample of 10 million from the 1 billion IG-1B images indicates that the dataset covers 192 geographic regions and represents various genders of people.)
>
> Regardless of the self-supervised model or training data, the self-supervised representations are poorly correlated with human uncertainty inherent to triplet odd-one-out probabilities. (If models are trained to reproduce labels created by humans or judgments/decisions made by humans, then it can be argued that models should also reflect human uncertainty when making predictions or judgments.)
> In addition, the representations result in worse accuracy than our baselines trained on CelebA, FairFace, and FFHQ. Finally, the representational space is poorly correlated with the human representational space of faces. We believe that this justifies the choice of baselines included in our main paper, i.e., models trained on tasks and labels that humans use to describe one another.
>
> Generally speaking, our results provide evidence of a gap between human representations of faces and the machine representations learned by the self-supervised models.  By distilling information about how humans represent and organize the visual world into machine representations, we can reduce the semantic gap in perception [1] and possibly increase the applicability of neural networks [2,3]. To reduce the semantic gap in perception, machine representations can be aligned with psychological embeddings [1-3]. One way is to train models to predict human similarity judgments [4-12], as we have done.
>
> Please let us know if this answers your question.
>
> ------------------------------
>
> [1] Wah, Catherine, et al. "Similarity comparisons for interactive fine-grained categorization." Proceedings of the IEEE Conference on Computer Vision and Pattern Recognition. 2014.\
> [2] Hsieh, Cheng-Kang, et al. "Collaborative metric learning." Proceedings of the 26th international conference on world wide web. 2017.\
> [3] Wilber, Michael, et al. "Learning concept embeddings with combined human-machine expertise." Proceedings of the IEEE International Conference on Computer Vision. 2015.\
> [4] Zhang, Richard, et al. "The unreasonable effectiveness of deep features as a perceptual metric." Proceedings of the IEEE conference on computer vision and pattern recognition. 2018.\
> [5] Lake, Brenden M., et al. "Deep Neural Networks Predict Category Typicality Ratings for Images." CogSci. 2015.\
> [6] Talebi, Hossein, and Peyman Milanfar. "NIMA: Neural image assessment." IEEE transactions on image processing 27.8 (2018): 3998-4011.\
> [7] Attarian, Maria, Brett D. Roads, and Michael C. Mozer. "Transforming neural network visual representations to predict human judgments of similarity." arXiv preprint arXiv:2010.06512 (2020).\
> [8] Sanders, Craig A., and Robert M. Nosofsky. "Training deep networks to construct a psychological feature space for a natural-object category domain." Computational Brain & Behavior 3.3 (2020): 229-251.\
> [9] Josephs, Emilie L., Martin N. Hebart, and Talia Konkle. "Emergent dimensions underlying human perception of the reachable world." Journal of Vision 21.9 (2021): 2154-2154.\
> [10] Hebart, Martin N., et al. "Revealing the multidimensional mental representations of natural objects underlying human similarity judgements." Nature human behaviour 4.11 (2020): 1173-1185.\
> [11] Zheng, Charles Y., et al. "Revealing interpretable object representations from human behavior." International Conference on Learning Representations. 2018.\
> [12] Dima, Diana C., Martin N. Hebart, and Leyla Isik. "A data-driven investigation of human action representations." bioRxiv (2022).

---

> > ### Author Response · Authors · 2022-11-12
> > **Response [2/2]**
> >
> > >Not sure whether the author is going to release the proposed dataset or code.
> >
> > The dataset will be released under a Creative Commons BY-NC-SA license and will be hosted on Zenodo without any access restrictions. In particular, we will release the following:
> > - 638,180 odd-one-out triplet judgments for training and validation, including all associated annotations (e.g., annotator demographic info)
> > - 24,060 odd-one-out judgments for Protocol 1 testing, including all associated annotations (e.g., annotator demographic info)
> > - 80,300 odd-one-out judgments for Protocol 2 testing, including all associated annotations (e.g., annotator demographic info)
> > - Topic labels provided by annotators for our learned dimensions, including all associated annotations (e.g., annotator demographic info)
> > - 8,800 judgments for the dimension rating task, including all associated annotations (e.g., annotator demographic info)
> >
> > The code to reproduce our experiments will be hosted on GitHub.
> >
> > We will post a WeTransfer link only visible to the reviewers with the code and dataset. We kindly request that the reviewers do not share the dataset. We plan on making the dataset publicly available on Zenodo at the camera ready stage, if our paper is successful.

---

### Official Review · Reviewer_L52N · 2022-11-01

**Confidence:** 5
**Clarity, Quality, Novelty And Reproducibility:** See above.
**Correctness:** 3
**Technical Novelty And Significance:** 3
**Empirical Novelty And Significance:** 2
**Recommendation:** 5

**Strength And Weaknesses:**

(+) Overall it's on an interesting and important topic.
(+) The paper is relatively well written and lots of details are provided in appendix.
(-) But I suggest to move important details to the main body, for example, how you estimate attributes to make intersectional groups to sample from FFHQ. It was not clear that the sample subset was balanced, which would be problematic because FFHQ is unbalanced.
(+) New dataset and applicable model

(-) My main reservation is validation. The paper shows many experiments, but everything from 3.1 - 3.5 is more of basic verification. Only 3.6 is about the demonstration of the utility of this dataset. As the primary contribution of this paper is a new dataset, I expect the authors show a lot more justifications on its usefulness than what was done in 3.6.

Most experiments (especially 3.6) focus on showing that the learned representation (each dimension) corresponds to demographic attributes such as race and gender. This is not surprising. This doesn't make the proposed dataset useful. If we just do k-means clustering, you will also discover these groupings automatically. So why do we need these new annotations? Is the primary goal of this paper to discover gender, race, age without annotating them directly on target data? If so, I think there are much cheaper ways. Why don't we just use a classifier to estimate these labels? While the paper argues that they want to avoid this, but the way the proposed representation is used and validated is pretty much the same (i.e. it has separate dimensions for these attributes). In other words, this demographic information is not "distributed", so I don't know if this is very different from other classifiers.

Also, 3.6 only uses CC and CFD. Both are very clean datasets. Will this work for in-the-wild dataset such as fairface?

Another question is that the representation still captures non-sensitive attributes such as smiling. The authors explicitly tried to avoid this. But it's still learned, which means the annotators still used expression.

Overall, I think the paper will benefit significantly from (1) having a clear discussion about how humans perceive and define similarity and (2) designing and performing evaluations aligned with that discussion. The current paper focuses only on gender, race, etc, and I don't see much benefit of this paper over explicit classifiers.


**Summary Of The Paper:**

This paper proposes a new face representation learned from similarity judgments and contributes a novel annotation dataset with 4900 faces  (a subset of FFHQ dataset). The main purpose of this representation is to allow us to measure sample diversity without explicit annotations. In this context, the diversity is defined mainly on demographic basis, i.e. race, gender, and so on. The annotation was made on sample triplets where the annotators were asked to choose a least-similar example out of 3 samples. The representation is learned by maximizing the similarity between the other two examples, with an annotator-specific weight vector, which is also learnable. Experiments show that the representation captures important demographic dimensions such as race, gender and also other ones such as smile, which the authors tried to avoid to learn.

**Summary Of The Review:**

I think this paper has a good potential and value. In my opinion, it's not fully demonstrated. In particular, Sec 3.6 needs to be expanded further. Competing baselines for this paper are not classifiers. I think they should be clustering, scaling, diversity sampling methods, etc.

In addition, I disagree with some statements like "we do not encode, reify, or propagate stereotypes". Even though the authors didn't ask the annotators to "use" stereotypes, there is no way to enforce it. The paper already shows that the annotators heavily relied on gender and race to define similarity. We categorize faces based on those dimensions. It's a known fact. (it may be interesting to show the amount of variance in annotations that can explained by gender age race)

Also, "Implicit in these annotations is the assumption that human annotators can consistently and objectively map observable characteristics to categorical labels." -> there's no assumption like this in prior work. It is well understood that annotators are biased and subjective. That's why these annotations are almost always obtained from multiple annotators. In any case, I don't clearly see how this proposed method can be safe from these issues because again, the evaluation shows the annotators did use the same signals. This can be clarified better or toned down. On the other hand, there's a clear benefit to not annotate these attributes (legal, privacy issues). I think it may be better to focus on that direction in motivation.

---

> ### Author Response · Authors · 2022-11-12
> **Response [1/10]**
>
> We would like to thank you for taking the time to review our paper and for providing detailed and helpful comments! We are particularly pleased that you think the paper is relatively well written and tackles an important topic. In addition, thank you for noting the effort put into our appendix, which includes our datasheet, and that we introduce a novel dataset and model.
>
> With regards to the concerns, our response is quite long as we wanted to make sure that we covered each point that you raised. Please let us know if you require any further information, or if anything is unclear.
>
> ---------------------------------
> >I suggest to move important details to the main body, for example, how you estimate attributes to make intersectional groups to sample from FFHQ.
>
> We agree that it will be helpful if we move details regarding the sampling of the stimuli set from the Appendix to Section 2.1. (Due to space limitations, we have added the details as a footnote in Section 2.1.)
>
> ---------------------------------
> >3.6 only uses CC and CFD. Both are very clean datasets. Will this work for in-the-wild dataset such as fairface?
>
> As our stimuli set was sampled from the in-the-wild FFHQ dataset, our model is best applied to high-quality face images. Note that CC, which we used in the experiments in Section 3.6, represents an image dataset of significantly lower quality face images compared to FFHQ. Note that we chose CC and CFD for the attribute disparity experiment, since they are the only datasets where the labels are actual ground-truths obtained directly from the image subjects, rather than estimated by annotators.
>
> We can update section 3.6 to include other datasets if you think it is important. (Section 3.5 already provides evidence of our model applied to other in-the-wild-datasets, namely FF, CA, COCO, and MIAP.)
>
> ---------------------------------
>
> >Another question is that the representation still captures non-sensitive attributes such as smiling. The authors explicitly tried to avoid this. But it's still learned, which means the annotators still used expression.
> We highlighted in Section 4 the apparent manifestation of two dimensions related to facial expression (smiling and neutral). We acknowledged that our set of annotators may not have followed our instruction to ignore facial expression. While this could be considered a flaw, our method was able to uncover these attributes since they explained some of the variation in the human judgments. Without our model the fact that annotators used these attributes would have otherwise remained undiscovered.
>
> Face datasets such as FairFace and CelebA do not provide any insight into which attributes their annotators used for categorization. (For instance, [1] found that an image subject's clothing influenced their racial categorization, i.e., low-status and high-status clothing increased the likelihood of being categorized as “Black” and “White”, respectively.) In contrast, our model, by design, provides insight into the dimensions used by our annotators when determining face similarity. Thus providing transparency.
>
> As an image subject can exhibit certain traits to a greater or lesser extent than others, even within the same subpopulation, the goal of our work is to learn a similarity function, which measures the similarity between two faces that is aligned with human perception.
>
> We never aimed to solely learn sensitive attributes. We only aimed to uncover attributes that could explain human behavior in the similarity task. Our instructions to ignore facial expression and accessories (e.g., hats, glasses, earrings) were given because we wanted to focus our proof-of-concept on intrinsic facial features. In future, when expanding our dataset, we plan on removing these instructions. In particular, extrinsic attributes such as accessories or lighting can be useful when attempting to find hidden confounders. Notably, [2] found that when moving in the StyleGAN2 latent space from dark-skinned men to dark-skinned women, earrings were added. Therefore, learning dimensions related to extrinsic factors could be helpful. However, this was beyond the scope of our current study.
>
> If you think that it would be helpful, we are happy to add this clarification to our paper or appendix.
>
> ---------------------------------
>
> [1] Freeman, Jonathan B., et al. "Looking the part: Social status cues shape race perception." PloS one 6.9 (2011): e25107.\
> [2] Balakrishnan, Guha, Yuanjun Xiong, Wei Xia, and Pietro Perona. 2020. “Towards Causal Benchmarking of Bias in Face Analysis Algorithms.” In The European Conference on Computer Vision (ECCV), 17.\

---

> > ### Author Response · Authors · 2022-11-12
> > **Response [2/10]**
> >
> > >I think the paper will benefit significantly from … having a clear discussion about how humans perceive
> >
> > **We are happy to provide more discussion on how humans perceive similarity either by adding to our related work section or the Appendix. For example, something along the following lines (note we have added to Appendix A some discussion on Similarity and Contextual Similarity; in addition, lines 80-84 introduce the topic of similarity):**
> >
> > Similarity represents a central theoretical construct in cognitive psychology [1,8,9], in particular as the human mind is conjectured by many scholars to have “a considerable investment in similarity” [1]. For example, human transfer learning depends heavily on similarity [3]. This is because, similarity, a type of comparison, permits people to make inferences on the basis of incomplete information.
> >
> > The process of comparing similarity hinges on the alignment of the attributes of the objects being compared [5]. The process of alignment is contended to be weighted such that the best alignment is chosen, where humans multiplicatively integrate over both matching and mismatching attributes to arrive at a single judgment [1]. Alignment involves looking at the attributes present in one object and measuring their fit in another object. In other words, when two objects are compared they mutually constrain the set of features that are activated or inferred in the human mind. For example, determining whether an object is red can depend on the object it is being compared to. As another example, comparing face A to faces B, C, and D will most likely involve different sets of features. That is, face A may be associated with different features when we vary the face it is being compared to, i.e., B, C, or D. The process of alignment is contended to be central to similarity comparisons. [4] argues that similarity is not a function of features, but that the features themselves are a function of the similarity comparison. This suggests that similarity is dynamic, where features are discovered and aligned based on what is being compared.
> >
> >
> > Notably, human perception of similarity can vary with respect to context [2,10]. For example, [2] found that a snake and raccoon were judged to be less similar when participants were not given any context. However, when the similarity comparison was contextualized, i.e., how similar are snakes and raccoons in the context of “pets” their similarity increased. The odd-one-out similarity task also provides context, for instance, it requires annotators to perform pairwise comparisons in order to determine the odd-one-out. By varying the context, e.g., face C in the triplet (A, B, C), we uncover different sets of features that contribute to the similarity and dissimilarity between A and B. This is important as there are an uncountable number of ways in which two objects may be similar or dissimilar [12]. Odd-one-out similarity judgments [11], or more generally contextual similarity comparisons, can be used to uncover the salient features that contribute to pairwise similarity. As highlighted in several works [1,8,9], context makes salient: (1) context-related properties; and (2) the extent to which objects being compared share context-related properties.
> >
> > ---------------------------------
> >
> > [1] Medin, Douglas L., Robert L. Goldstone, and Dedre Gentner. "Respects for similarity." Psychological review 100.2 (1993): 254.\
> > [2] Barsalou, Lawrence. "The instability of graded structure: Implications for the nature of concepts." (1987).\
> > [3] Thorndike, L. Edward. "Human learning." The Journal of Nervous and Mental Disease 75.5 (1932): 589.\
> > [4] Shanon, Benny. "On the similarity of features." New ideas in psychology 6.3 (1988): 307-321.\
> > [5] Markman, Arthur B. "Structural alignment in similarity and difference judgments." Psychonomic Bulletin & Review 3.2 (1996): 227-230.\
> > [6] Medin, Douglas L., and Marguerite M. Schaffer. "Context theory of classification learning." Psychological review 85.3 (1978): 207.\
> > [7] Nosofsky, Robert M. "Exemplar-based accounts of relations between classification, recognition, and typicality." Journal of Experimental Psychology: learning, memory, and cognition 14.4 (1988): 700.\
> > [8] Markman, Arthur B., and Dedre Gentner. "Nonintentional similarity processing." The new unconscious (2005): 107-137.\
> > [9] Goodman, Nelson. "Seven strictures on similarity." (1972).\
> > [10] Roth, Emilie M., and Edward J. Shoben. "The effect of context on the structure of categories." Cognitive psychology 15.3 (1983): 346-378.\
> > [11] Hebart, Martin N., et al. "Revealing the multidimensional mental representations of natural objects underlying human similarity judgements." Nature human behaviour 4.11 (2020): 1173-1185.\
> > [12] Love, Bradley C., and Brett D. Roads. "Similarity as a Window on the Dimensions of Object Representation." Trends in Cognitive Sciences 25.2 (2021): 94-96.

---

> > > ### Author Response · Authors · 2022-11-12
> > > **Response [3/10]**
> > >
> > > >I think the paper will benefit significantly from … designing and performing evaluations aligned with that discussion.
> > > >The paper shows many experiments, but everything from 3.1 - 3.5 is more of basic verification. Only 3.6 is about the demonstration of the utility of this dataset.
> > > >As the primary contribution of this paper is a new dataset, I expect the authors show a lot more justifications on its usefulness than what was done in 3.6.
> > >
> > > **We believe that our experiments are aligned with the evaluation of human perception of face similarity. Moreover, we show a multiplicity of uses of our model’s learned embedding space:**
> > >
> > > - First, we show that we can predict human judgments of similarity for novel triplets better than baseline representations (Section 3.1). This shows that training on categorical labels and representations induced by learning on categorical attributes fail to fully reflect how humans determine similarity. As mentioned in our introduction, the standard approach is to categorize people using demographic attributes labels. Many equate diversity with parity across the subgroup distributions. However, having a balanced dataset wrt to subgroups does not mean that the subgroups are, or the dataset is, diverse. For example, balancing based on subgroup labels has been shown to be an insufficient measure of diversity when mitigating bias amplification.
> > > - Second, the uncertainty inherent to human triplet odd-one-out probabilities has a higher correlation with our model-generated triplet odd-one-out probabilities than the baselines (Section 3.1). In addition, our model results in similarity matrices that have a higher correlation with similarity matrices generated from human judgments than the baselines. This provides evidence of a gap between current machine representations and human representations.
> > > - Third, we show that the importance distinct annotators place on each of the embedding dimensions when determining similarity differs (Section 3.2). Further, we show that the annotator importance vectors can be grouped based on the sociocultural factors of the annotators. Thus, we show that human annotators perceive similarity differently based on their sociocultural background.  To our knowledge, our work is the first piece of empirical research in computer vision highlighting annotator bias wrt sociocultural factors. This has consequences for all dataset creators, i.e., by failing to model annotators as a contributing factor in what is learned by models we are ignoring an obvious source of bias. Therefore, if we were to deploy a model trained on a dataset that was labeled entirely by, e.g., Americans with Western European ancestry, then our model will view faces from this narrow perspective. This is not inclusive and potentially harmful. Moreover, our conditional decision-making framework can be used for other tasks, as long as each data instance is associated with the annotator who generated the label or judgment.
> > > - Fourth, we show that continuous attribute labels can be collected for novel faces using image grids created using our interpretable model dimensions (Section 3.3). As we mention in the paper, these human-interpretable grids can be generated using other dimensions from other models. By highlighting that we can generate accurate face representations by showing participants image grids, by having them place novel images on the grid, we can avoid asking annotators to label people altogether. This is also useful if we only care about a single dimension and wish to obtain a label for a new image from a human annotator.
> > > - Finally, we show that a subset of our dimensions are correlated with human judgments of face prototypicality (Section 3.4). This is an interesting finding, since our similarity task does not present annotators with labels or prototypes. Therefore, prototypicality naturally surfaces from human similarity judgments, i.e., the judgments result in an ordering of the faces along dimensions that explain the variation in human judgments. (Note that this experiment focused on gender and U.S. race categories, because they were the only attributes represented in Chicago Face Database, and no other resource exists.)
> > > In addition Section 3.5 shows that the individual dimensions in our learned face embedding space can act as attribute classifiers. The combined findings in Section 3.4 and 3.5 show that we do not need to ask annotators to label faces in order to learn these attributes.
> > >
> > >
> > > ---------------------------------
> > >
> > > [1] Wang, Tianlu, et al. "Balanced datasets are not enough: Estimating and mitigating gender bias in deep image representations." Proceedings of the IEEE/CVF International Conference on Computer Vision. 2019.

---

> > > > ### Author Response · Authors · 2022-11-12
> > > > **Response [4/10]**
> > > >
> > > > >Is the primary goal of this paper to discover gender, race, age without annotating them directly on target data? If so, I think there are much cheaper ways. Why don't we just use a classifier to estimate these labels? While the paper argues that they want to avoid this, but the way the proposed representation is used and validated is pretty much the same (i.e. it has separate dimensions for these attributes). In other words, this demographic information is not "distributed", so I don't know if this is very different from other classifiers.
> > > >
> > > > As an image subject can exhibit certain traits to a greater or lesser extent than others, even within the same subpopulation, the goal of our work is to learn a similarity function, which measures the similarity between two faces that is aligned with human perception. Alignment with human perception relates to the use of attributes that humans use when determining similarity. Therefore, we aim to learn a human-interpretable face embedding space that is aligned with the human mental representational space.
> > > >
> > > > We validate on similar tasks to classification models, because we want to show that it’s possible to learn similar information in a way that reduces harm to the image subjects—i.e., our odd-one-out task does not encode, reify, or propagate categorization systems beyond their cultural context. We are able to learn discriminative attribute classifiers from similarity judgments alone without collecting harmful labels. Moreover, we show that feature spaces learned by demographic classifiers are not aligned with the way humans represent faces in their minds. As similarity is inversely connected to diversity, this implies that explicit face attribute labels, which are regularly used to measure dataset diversity, do not fully account for the dimensions used by humans when determining similarity or diversity.
> > > >
> > > > In order to convince others that they do not need to collect explicit face attribute labels, in our work we show that attributes learned through similarity judgments give competitive results on common tasks, and that our embeddings are closer to human mental representations of faces.
> > > >
> > > > Regarding the distribution of demographic information. Our dataset represents 0.003% of all possible triplets that can be sampled from 4,921 images. To go one step further, triplets should be sampled selectively so as to learn other face-varying dimensions. That is, our dataset can be extended using active learning approaches, where we focus on triplets that have high prediction uncertainty. For instance, this could correspond to triplets composed of faces that are close in embedding space and therefore similarity judgments are unlikely to be based on gender, skin color, or ethnicity, but rather hair color, eye color, etc. Using our annotator labels, uncertainty quantification can also incorporate the annotator that will perform a judgment. Active learning approaches typically assume a single oracle, however as we have shown different oracles do not place the same importance on face attributes when determining similarity. Therefore, our dataset will enable future researchers to study active learning in a realistic setting where there are multiple oracles from varying backgrounds.
> > > >
> > > > If you think it would be helpful, we can add a discussion point on the implications of our results on active learning, as well as how our dataset can be used by researchers working on active learning.

---

> > > > > ### Author Response · Authors · 2022-11-12
> > > > > **Response [5/10]**
> > > > >
> > > > > >Is the primary goal of this paper to discover gender, race, age without annotating them directly on target data? If so, I think there are much cheaper ways.
> > > > > >The current paper focuses only on gender, race, etc, and I don't see much benefit of this paper over explicit classifiers.
> > > > > The benefits of our dataset and model over explicit classifiers that are trained on semantically labeled data are as follows:
> > > > > - On the one hand, semantic labels obtained directly from human subjects present privacy concerns, in particular in the current climate of privacy regulation and data minimization. On the other hand, semantic labels inferred by models can be harmful when considering that data subjects can make data access requests—i.e., when inferred labels do not correspond to a subject’s self-identified labels this can cause psychological harm to the subject. Our dataset does not contain any labels, nor do we label images using our model. We learn face representations that can be interpreted by humans, which permit the measurement of pairwise face similarity.
> > > > > - Inferred semantic labels can be more costly than odd-one-out judgments, since they typically seek label consensus. Moreover, as we increase the number of possible categories for an attribute, we also increase the amount of time an annotator is required to spend on labeling an image. When label consensus cannot be reached for an image, dataset authors such as the FairFace authors discard the image. This likely erases, e.g., multi-ethnic, trans individuals, as well as others that do not conform to a stereotype. Our dataset does not exclude people because they do not conform to stereotypes, moreover our learned representations are continuous which permits the representation of people who do not fit neatly into rigid systems of categorization.
> > > > > - Asking annotators to label images requires researchers to predetermine a set of attributes that they want labeled, however, we do not know a priori whether a label even applies to any of the data points. Furthermore, researchers are often accused of using problematic label schemas that are not only poorly formatted, but also include derogatory and offensive labels [1,2,3,4]. This is especially harmful when the labels are culturally and politically rooted [5].
> > > > >
> > > > > ---------------------------------
> > > > >
> > > > > [1] Crawford, Kate, and Trevor Paglen. "Excavating AI: The politics of images in machine learning training sets." Ai & Society (2021): 1-12.\
> > > > > [2] Birhane, Abeba, and Vinay Uday Prabhu. "Large image datasets: A pyrrhic win for computer vision?." 2021 IEEE Winter Conference on Applications of Computer Vision (WACV). IEEE, 2021.\
> > > > > [3] Hanley, Margot, et al. "An ethical highlighter for people-centric dataset creation." Navigating the Broader Impacts of AI Research Workshop at NeurIPS. 2020.\
> > > > > [4] Koch, Bernard, et al. "Reduced, Reused and Recycled: The Life of a Dataset in Machine Learning Research." Thirty-fifth Conference on Neural Information Processing Systems Datasets and Benchmarks Track (Round 2). 2021.\
> > > > > [5] Khan, Zaid, and Yun Fu. "One label, one billion faces: Usage and consistency of racial categories in computer vision." Proceedings of the 2021 acm conference on fairness, accountability, and transparency. 2021.

---

> > > > > > ### Author Response · Authors · 2022-11-12
> > > > > > **Response [6/10]**
> > > > > >
> > > > > > >In particular, Sec 3.6 needs to be expanded further. Competing baselines for this paper are not classifiers. I think they should be clustering, scaling, diversity sampling methods, etc.
> > > > > >
> > > > > > If possible, please can you elaborate on why competing baselines should not be classifiers. In particular, the relevance of clustering, scaling, and diversity sampling methods for measuring attribute disparity which is the topic of Section 3.6. Any direction on this would be very helpful, thank you!
> > > > > >
> > > > > > ---------------------------------
> > > > > >
> > > > > > >In addition, I disagree with some statements like "we do not encode, reify, or propagate stereotypes". Even though the authors didn't ask the annotators to "use" stereotypes, there is no way to enforce it.
> > > > > >
> > > > > >
> > > > > > Our task is completely open-ended. We do not constrain participants to use any particular attribute when determining similarity. Therefore, by design, the odd-one-out task does not rely on stereotypes. To clarify, in other computer vision datasets, annotators are given semantic categories to label images as belonging to. In addition, annotators may be shown example images (i.e., prototypes) for each semantic category and/or given descriptions of the category. The example descriptions and/or prototype images result in stereotypical annotations, since they require annotators to compare the “fit” of an unlabeled image to category labels, descriptions, or prototypes.
> > > > > >
> > > > > >
> > > > > > If we focus on social constructs, which are relevant when dealing with human-centric data:
> > > > > > 1. semantic category labels represent **encodings** of social constructs;
> > > > > > 2. requesting annotators to use semantic category labels **reifies** social constructs, i.e., makes the social constructs real, despite their contentious scientific validity;
> > > > > > 3. asking annotators external to the society from which a social construct originates **propagates** the construct beyond its cultural and societal context.
> > > > > >
> > > > > > The similarity task given to our annotators does not encode, reify, or propagate any attribute or social construct, since we do not prime the annotators with labels, descriptions, or prototypes. This does not mean that our similarity task is free from bias, as evidenced by the increased accuracy of our conditional model and annotator bias analysis. Our statement was only related to the use of contentious categories in annotation tasks, especially categories that are culturally and politically rooted. Our statement aligns with [1]’s study.
> > > > > >
> > > > > > We are happy to provide this clarification within the paper or appendix. However, if you do not agree with this, please let us know. **Note that we have updated lines 39-41 to reflect the fact that our statement is about categorical schemas.**
> > > > > >
> > > > > > ---------------------------------
> > > > > >
> > > > > > [1] Khan, Zaid, and Yun Fu. "One label, one billion faces: Usage and consistency of racial categories in computer vision." Proceedings of the 2021 acm conference on fairness, accountability, and transparency. 2021.

---

> > > > > > > ### Author Response · Authors · 2022-11-12
> > > > > > > **Response [7/10]**
> > > > > > >
> > > > > > > >The paper already shows that the annotators heavily relied on gender and race to define similarity. We categorize faces based on those dimensions. It's a known fact. (it may be interesting to show the amount of variance in annotations that can explained by gender age race)
> > > > > > >
> > > > > > > **We have added to Appendix F results where we use the labels of the face images in triplets to determine the odd-one-out.** As gender, race, and age categorical labels can only explain odd-one-out behavior for triplets where two of the three faces share an attribute (whereas the third face does not), we perform these experiments on subsets of the data that satisfy these conditions. For instance, for gender, we consider triplets where there are two gender expression labels (Male, Female) present within a triplet. Note: gender, race, and age cannot explain human behavior for triplets where all three faces share the same attribute(s) or all three faces share zero attributes.
> > > > > > >
> > > > > > > **We also include an additional experiment in Section 3.1, where we estimate the number of dimensions required to represent the human mental representation space of faces.** The results show that humans utilize a larger number of dimensions to represent the global similarity structure of faces (i.e., 15-22) than for predicting individual odd-one-out judgments (i.e., 6-13).

---

> > > > > > > > ### Author Response · Authors · 2022-11-12
> > > > > > > > **Response [8/10]**
> > > > > > > >
> > > > > > > > >Also, "Implicit in these annotations is the assumption that human annotators can consistently and objectively map observable characteristics to categorical labels." -> there's no assumption like this in prior work. It is well understood that annotators are biased and subjective. That's why these annotations are almost always obtained from multiple annotators. In any case, I don't clearly see how this proposed method can be safe from these issues because again, the evaluation shows the annotators did use the same signals. This can be clarified better or toned down. On the other hand, there's a clear benefit to not annotate these attributes (legal, privacy issues). I think it may be better to focus on that direction in motivation.
> > > > > > > >
> > > > > > > >
> > > > > > > > Thank you for your advice! We have removed the quoted sentence and updated the text, please **see lines 39-41**. We agree that the legal and privacy issues are important and worth focusing on.
> > > > > > > >
> > > > > > > > To address your concern, we are happy to include something along the following lines in the main paper or Appendix (at present we have included the following in **Appendix A: Data Protection and Privacy**):
> > > > > > > >
> > > > > > > > Many practitioners have said that they avoid collecting special/sensitive category data (e.g., race), which is subject to stronger restrictions, so as to avoid potential privacy violations [1]. Privacy and legal counsel typically advise against collecting demographic information even if it is for fairness purposes [1], following the principle of data minimization—i.e., only collect what you need. However, if we are unable to collect demographic and personal information (as well as other attributes) for developing fairer and more ethical AI systems, then we require creative solutions to ensure that datasets have some level of diversity and are representative. Our dataset and model offers such a creative solution: (1) our dataset does not contain any attribute labels; and (2) our model does not label faces.
> > > > > > > >
> > > > > > > >
> > > > > > > > While labels can be inferred using attribute classifiers, there are concerns with this, which have been voiced by many practitioners [1]. Data access request rights (e.g., via GDPR, CCPA, PIPL) may force data holders into revealing information held about data subjects, including inferred information. Due to this, practitioners are more cautious about inferring labels about people [1], especially as mismatches between inferred and self-identified race, gender, or other sensitive and nonsensitive attributes can induce psychological distress [2] by invalidating an individual's self-image and identity [3]. Our method instead implicitly learns face-varying axes from similarity judgments, altogether avoiding any explicit labeling of faces. We show that the axes are meaningful and human-interpretable through the dimension labeling and dimension rating human experiments. Our model (i.e., similarity function) provides dataset users with a new type of diversity measure that coincides with human perception, which can be plugged into any metric.
> > > > > > > >
> > > > > > > >
> > > > > > > > ---------------------------------
> > > > > > > >
> > > > > > > > [1] Andrus, McKane, et al. "What We Can't Measure, We Can't Understand: Challenges to Demographic Data Procurement in the Pursuit of Fairness." Proceedings of the 2021 ACM Conference on Fairness, Accountability, and Transparency. 2021.\
> > > > > > > > [2] Campbell, Mary E., and Lisa Troyer. "The implications of racial misclassification by observers." American Sociological Review 72.5 (2007): 750-765.\
> > > > > > > > [3] Roth, Wendy D. "The multiple dimensions of race." Ethnic and Racial Studies 39.8 (2016): 1310-1338.

---

> > > > > > > > > ### Author Response · Authors · 2022-11-12
> > > > > > > > > **Response [9/10]**
> > > > > > > > >
> > > > > > > > > **Regarding annotator positionality:** \
> > > > > > > > > (We have updated our related work section on Annotator Bias to reflect the following response; **please see lines 98-104**)
> > > > > > > > >
> > > > > > > > > The role of the annotator has only recently entered into discourse [1,2], albeit predominantly in the context of natural language processing [3,4,5,6,7,8]. Recent empirical studies have shown that “relatively little attention is given or documented about annotator positionality—how annotator social identity shapes their understanding of the world” [10,11,12].
> > > > > > > > >
> > > > > > > > > In order to mitigate bias, one must first measure bias [9]. Looking into the disciplinary values in computer vision dataset development, it has been noted that **from 113 computer vision datasets surveyed** [10]: “dataset creators tended to omit critical details regarding who was performing the annotation tasks. For example, **only five publications provided any demographic information regarding who annotated the data instances**”.
> > > > > > > > >
> > > > > > > > > Unlike previous works, we record which annotator performed which annotation and also record their demographic attributes. Our dataset permits users to study per-annotator bias (i.e., annotators as individuals) as well as biases related to an annotator’s background (i.e., annotator groups). Furthermore, our model of conditional decision-making provides a mechanism to study the importance different annotators place on face-varying dimensions. Thus we acknowledge that our dataset and learned embedding space represent a view from somewhere (i.e., our set of annotators), as opposed to the aforementioned computer vision datasets that take a view from nowhere [10].
> > > > > > > > >
> > > > > > > > > ---------------------------------
> > > > > > > > >
> > > > > > > > > [1] Chen, Yunliang, and Jungseock Joo. "Understanding and mitigating annotation bias in facial expression recognition." Proceedings of the IEEE/CVF International Conference on Computer Vision. 2021.\
> > > > > > > > > [2] Zhao, Dora, Angelina Wang, and Olga Russakovsky. "Understanding and evaluating racial biases in image captioning." Proceedings of the IEEE/CVF International Conference on Computer Vision. 2021.\
> > > > > > > > > [3] Al Kuwatly, Hala, Maximilian Wich, and Georg Groh. "Identifying and measuring annotator bias based on annotators’ demographic characteristics." Proceedings of the Fourth Workshop on Online Abuse and Harms. 2020.\
> > > > > > > > > [4] Waseem, Zeerak. "Are you a racist or am i seeing things? annotator influence on hate speech detection on twitter." Proceedings of the first workshop on NLP and computational social science. 2016.\
> > > > > > > > > [5] Wich, Maximilian, Hala Al Kuwatly, and Georg Groh. "Investigating annotator bias with a graph-based approach." Proceedings of the fourth workshop on online abuse and harms. 2020.\
> > > > > > > > > [6] Binns, Reuben, et al. "Like trainer, like bot? Inheritance of bias in algorithmic content moderation." International conference on social informatics. Springer, Cham, 2017.\
> > > > > > > > > [7] Salminen, Joni, et al. "Online hate interpretation varies by country, but more by individual: A statistical analysis using crowdsourced ratings." 2018 fifth international conference on social networks analysis, management and security (snams). IEEE, 2018.\
> > > > > > > > > [8] Geva, Mor, Yoav Goldberg, and Jonathan Berant. "Are we modeling the task or the annotator? an investigation of annotator bias in natural language understanding datasets." arXiv preprint arXiv:1908.07898 (2019).\
> > > > > > > > > [9] Le Quy, Tai, et al. "A survey on datasets for fairness‐aware machine learning." Wiley Interdisciplinary Reviews: Data Mining and Knowledge Discovery (2022): e1452.\
> > > > > > > > > [10] Scheuerman, Morgan Klaus, Alex Hanna, and Emily Denton. "Do datasets have politics? Disciplinary values in computer vision dataset development." Proceedings of the ACM on Human-Computer Interaction 5.CSCW2 (2021): 1-37.\
> > > > > > > > > [11] Denton, Emily, et al. "Whose ground truth? accounting for individual and collective identities underlying dataset annotation." arXiv preprint arXiv:2112.04554 (2021).\
> > > > > > > > > [12] Geiger, R. Stuart, et al. "Garbage in, garbage out? Do machine learning application papers in social computing report where human-labeled training data comes from?." Proceedings of the 2020 Conference on Fairness, Accountability, and Transparency. 2020.

---

> > > > > > > > > > ### Author Response · Authors · 2022-11-12
> > > > > > > > > > **Response [10/10]**
> > > > > > > > > >
> > > > > > > > > > >Most experiments (especially 3.6) focus on showing that the learned representation (each dimension) corresponds to demographic attributes such as race and gender. This is not surprising. This doesn't make the proposed dataset useful. If we just do k-means clustering, you will also discover these groupings automatically. So why do we need these new annotations?
> > > > > > > > > >
> > > > > > > > > > If possible, please can you elaborate on which inputs k-means clustering is performed on. If you are referring to clustering our learned representations, which you have stated would result in the automated discovery of meaningful groups, then this would show that our annotations are useful. That is, learning on our annotations results in a feature space that “groups” faces based on meaningful face attributes.
> > > > > > > > > >
> > > > > > > > > > To clarify, we are not trying to replicate demographic attributes but instead go beyond them; the reason we still provide comparisons with demographic attribute classifiers is to show that we are learning meaningful attributes. In particular, as if we did not, then some might think that the signal in FAX might be too weak and that nothing meaningful can be learned from the similarity judgments. Therefore, the goal was to show that is not the case.

---

> > > > > > > > > > > ### Comment · Reviewer_L52N · 2022-11-18
> > > > > > > > > > > **Update**
> > > > > > > > > > >
> > > > > > > > > > > I'd like to thank the authors for their extensive elaboration! I read it very carefully. Many of my questions were well addressed or at least better clarified. These are my main comments after reading the response.
> > > > > > > > > > >
> > > > > > > > > > > 1. I still disagree with "we do not encode, reify, or propagate stereotypes, since we do not use categorical labels, label descriptions, or prototypical imagery." The annotators were not asked to use gender/race. This doesn't mean the annotators won't use stereotypes. I think there's misunderstanding between us about what a stereotype means. My definition is broader. There are many machine learning datasets where annotators were not asked to rate gender/race, but the resulting models are still biased. This is because the annotators will use and project their own perception and stereotypes about the world in the annotations on target variables, whether they are image class, hate speech, etc. Similarity is not an exception. For example, a triplet may contain White-Man, White-Woman, Black-Man. Some people may choose White-Woman if they think gender difference is bigger than race, other may choose Black-Man. This is a stereotype on demographic distance. Another example, people will still infer gender from gender-neutral face (maybe from clothing) and can use this categorization when assessing the similarity. Again this is a stereotype on people's appearance.
> > > > > > > > > > >
> > > > > > > > > > > As long as we ask subjective perceptual judgment, there is no way to make these annotations completely independent from our stereotypes. I don't even think this argument is important in this paper. The advantage of this paper is that they didn't directly ask race/gender. I think that's good enough, and no reason to say much beyond it.
> > > > > > > > > > >
> > > > > > > > > > > 2. I made a few suggestions on the experiment in 3.6. Mainly this is about measuring dataset diversity, which is an important topic. I suggested the authors consider other baselines because those are used for the task of measuring dataset diversity (clustering, metric learning, etc). I still suggest the same thing. Also the authors said they could add more results using other datasets. Yes, I think they will be very helpful. Clustering on the features learned from your annotations will be great to show, but also on other features from other models can be good baselines.
> > > > > > > > > > >
> > > > > > > > > > > 3. I appreciate many other insightful responses. I do think some of these can go into the main paper if space is permitted.

---

> > > > > > > > > > > > ### Author Response · Authors · 2022-11-19
> > > > > > > > > > > > **Update response**
> > > > > > > > > > > >
> > > > > > > > > > > > Thank you for taking the time to read our responses!
> > > > > > > > > > > >
> > > > > > > > > > > > --------------------------------
> > > > > > > > > > > > **(1)** We agree that if this is a point of contention, then it is better that we focus on the various other reasons that one may want to avoid categorizing people or collecting labels. Therefore, based on your reply, we have removed from the paper mentions of categorization schemas encoding, reifying, and propagating stereotypes. For example, in the introduction lines 40-42 we have focused on data access request rights through e.g. GDPR, CCPA, and PIPL.
> > > > > > > > > > > >
> > > > > > > > > > > > --------------------------------
> > > > > > > > > > > >
> > > > > > > > > > > > **(2)** We have extended Tables 4 and 5, which now report results for the following datasets (which vary in image quality, pose, occlusions, etc): Casual Conversations, FFHQ, Chicago Face Database, COCO 2014 validation, OpenImages MIAP, and CelebA. In particular, as you were interested in seeing performance on datasets that are *less* constrained. We hope that the extension to Table 5 in the **comparative dataset diversity auditing section** is satisfactory.
> > > > > > > > > > > >
> > > > > > > > > > > > Despite the time constraint wrt to updating the paper prior to the deadline, we were also able to add a new table (Table 6). Table 6 provides normalized mutual information scores for **semantic clustering** on a variety of datasets. In particular, we utilized the underlying number of subgroups in a dataset and performed $k$-means clustering with $k$ equal to the number of ground-truth subgroups. We find that our embeddings are mostly competitive, despite never having been trained on semantic labels.
> > > > > > > > > > > >
> > > > > > > > > > > >
> > > > > > > > > > > > Based on the results in Table 1, the results in Table 6 are not too surprising. In particular, as we evidenced a gap between the similarity structure of embedding spaces induced by learning to predict face identity/face attributes and our representations which were induced by learning to predict similarity judgments (Table 1). Therefore, clustering embeddings and measuring their "purity" wrt to semantic labels represents a scenario where semantically trained representations are most useful.
> > > > > > > > > > > >
> > > > > > > > > > > > --------------------------------
> > > > > > > > > > > >
> > > > > > > > > > > > **(3)** Thank you for your kind words! We have updated the paper to reflect the majority of your comments and our responses to your comments, where possible. The appendix has also been extended.
> > > > > > > > > > > >
> > > > > > > > > > > > --------------------------------
> > > > > > > > > > > >
> > > > > > > > > > > > Thanks again for your initial review and recent response. We appreciate it. Please feel free to provide any additional feedback as you see fit.

---

### Author Response · Authors · 2022-11-29
**Discussion Period 2**

Dear Reviewers,

Please let us know if you have any additional comments or require further clarification on any of the points raised so that we can try to address them before the end of the discussion period on December 12.

Thank you in advance for your help.

---

### Decision · Program_Chairs · 2023-01-20

**Decision:**

Accept: poster

**Justification For Why Not Higher Score:**

This is mainly a dataset paper. The proposed method and results are relatively preliminary

**Justification For Why Not Lower Score:**

The new dataset of 638,180 human judgments of face similarity is worth sharing at ICLR

**Metareview: Summary, Strengths And Weaknesses:**

This paper proposes an implicit learning method for continuous face-varying dimensions with weak human annotations. The proposed topic of human-centric face representation is interesting, and the dataset introduced in this paper will contribute to the community, e.g. the result can be used in auditing datasets for diversity; FAX, a novel dataset of 638,180 face similarity judgments over 4,921 faces, can be of interest for researchers from multiple disciplines.

**Note From Pc:**

if the above contains the word "oral" or "spotlight" please see: "oral" presentation means -> notable-top-5% and "spotlight" means -> notable-top-25%. As stated in our emails, we are disassociating presentation type from AC recommendations

**Summary Of Ac-Reviewer Meeting:**

NA, 2 out of 3 reviewers accept the paper and the third review (score 5) has concern about validation but authors have addressed them rigorously. In any case the key contribution is more on the dataset, and to extend the research in human-centric face representation, such as similarity judgments in human perception